# Efficient Bayesian Learning Curve Extrapolation using Prior-Data Fitted Networks

**Steven Adriaensen**[*]
Machine Learning Lab
University of Freiburg
adriaens@cs.uni-freiburg.de

**Herilalaina Rakotoarison**[*]
Machine Learning Lab
University of Freiburg
rakotoah@cs.uni-freiburg.de

**Samuel Müller**
Machine Learning Lab
University of Freiburg
muellesa@cs.uni-freiburg.de

**Frank Hutter**
Machine Learning Lab
University of Freiburg
fh@cs.uni-freiburg.de

## Abstract

Learning curve extrapolation aims to predict model performance in later epochs of training, based on the performance in earlier epochs. In this work, we argue that, while the inherent uncertainty in the extrapolation of learning curves warrants a Bayesian approach, existing methods are (i) overly restrictive, and/or (ii) computationally expensive. We describe the first application of prior-data fitted neural networks (PFNs) in this context. A PFN is a transformer, pre-trained on data generated from a prior, to perform approximate Bayesian inference in a single forward pass. We propose `LC-PFN`, a PFN trained to extrapolate artificial right-censored learning curves generated from a parametric prior proposed in prior art using MCMC. We demonstrate that `LC-PFN` can approximate the posterior predictive distribution over learning curves more accurately than MCMC, while being over 10 000 times faster. We also show that the same `LC-PFN` achieves competitive performance extrapolating a total of 20 000 real learning curves from four learning curve benchmarks (`LCBench`, `NAS-Bench-201`, `Taskset`, and `PD1`) that stem from training a wide range of model architectures (MLPs, CNNs, RNNs, and Transformers) on 53 different datasets with varying input modalities (tabular, image, text, and protein data). Finally, we investigate its potential in the context of model selection and find that a simple `LC-PFN` based predictive early stopping criterion obtains 2 - 6× speed-ups on 45 of these datasets, at virtually no overhead.

## 1  Introduction

Learning curve extrapolation [Mohr and van Rijn, 2022] aims to predict how much a machine learning model will improve with more training, e.g., to determine how much more training data to collect [Cortes et al., 1993, Frey and Fisher, 1999, Leite and Brazdil, 2004, Kolachina et al., 2012], or to define an early stopping criterion in online learning [Yao et al., 2007]. Learning curve extrapolation has recently been widely studied to speed up automated machine learning (AutoML) and hyperparameter optimization (HPO) of deep neural networks, by discarding non-promising configurations early [Swersky et al., 2014, Domhan et al., 2015, Klein et al., 2017, Baker et al., 2017, Chandrashekaran and Lane, 2017, Gargiani et al., 2019, Wistuba et al., 2022].

---

[*] Equal Contribution.

37th Conference on Neural Information Processing Systems (NeurIPS 2023).

Despite these efforts, learning curve extrapolation is not yet widely adopted in practice, e.g., state-of-the-art multi-fidelity hyperparameter optimization techniques, such as BOHB [Falkner et al., 2018], still rely on successive halving [Li et al., 2017], i.e., the crude heuristic that learning curves mostly do not cross each other.

One reason for this is that, while many learning curves are well-behaved, some exhibit chaotic behavior and are intrinsically difficult to predict accurately [Choi et al., 2018]. In this setting, Bayesian approaches [Swersky et al., 2014, Domhan et al., 2015, Klein et al., 2017, Wistuba et al., 2022], which also quantify the reliability of their extrapolation, show great potential. However, existing methods for Bayesian inference either (i) put strong restrictions on the prior, and are incapable of modeling the variable nature of learning curves, or (ii) are too computationally expensive, limiting their practical applicability. Furthermore, most of this related work focuses on demonstrating the potential that learning curve extrapolation has to accelerate downstream AutoML/HPO tasks, yet fails to fully investigate the quality of the extrapolations themselves, and to quantify the approach's ability to handle the heterogeneity of real-world learning curves, e.g., varying performance metrics, curve shapes, divergence, heteroscedastic noise, etc.

In this work, we investigate the potential of learning curve extrapolation using prior-data fitted networks (PFNs), a meta-learned approximate Bayesian inference method recently proposed by Müller et al. [2022]. PFNs combine great flexibility with efficient and accurate approximation of the posterior predictive distribution (PPD) in a single forward pass of a transformer [Vaswani et al., 2017] trained on artificial data from the prior only. As PFNs are a promising alternative to Markov Chain Monte Carlo (MCMC) for approximating Bayesian inference, we compare our approach (`LC-PFN`) to the MCMC approach for learning curve extrapolation of Domhan et al. [2015], taking into account both the quality and the cost of PPD approximation.

In summary, our contributions are as follow:

- We are the first to apply PFNs to an extrapolation task, introducing `LC-PFN`, the first PFN for learning curve extrapolation.

- We demonstrate that `LC-PFN` can be more than $10\,000\times$ faster than MCMC while still yielding better probabilistic extrapolations.

- We show that `LC-PFN` does not only yield better probabilistic extrapolations on prior samples, but also on real learning curves of a wide range of architectures (MLPs, CNNs, RNNs, Transformers) on varying input modalities (tabular, image, text and protein data).

- We demonstrate the practical usefulness of `LC-PFN` to construct an early stopping criterion that achieves $2\text{-}6\times$ speedups over baselines.

- To facilitate reproducibility and allow others to build on our work, we open-source all code, data, and models used in our experiments at `https://github.com/automl/lcpfn`.

## 2   Related work

Learning curves and how to use them for decision-making has been an active research area, as recently surveyed by Mohr and van Rijn [2022]. Most related work considers point estimates of the curve or a specific property thereof [Cortes et al., 1993, Frey and Fisher, 1999, Kolachina et al., 2012, Baker et al., 2017, Kaplan et al., 2020], or follows a non-Bayesian approach to quantify uncertainty [Chandrashekaran and Lane, 2017, Gargiani et al., 2019].

Only a few works have explored Bayesian learning curve extrapolation. For example, the Freeze-Thaw Bayesian optimization method Swersky et al. [2014] used a Gaussian process (GP) as a joint model of learning curves and hyperparameters to decide what learning run to continue for a few epochs (or whether to start a new one). The model is then dynamically updated to fit the partial learning curve data. Training data grows quickly since each performance observation of the curve is treated as a datapoint, making exact GPs intractable, and thus the work relies on approximate GPs. Furthermore, their approach makes strong (prior) assumptions. On top of the standard GP assumptions, they used a specialized kernel assuming exponential growth to improve extrapolation. Domhan et al. [2015] proposed a less restrictive parametric prior (see Section 3.2 for more details) and used the gradient-free MCMC method from Foreman-Mackey et al. [2013] as approximate inference method. While MCMC is a very general approach, it can be sensitive to its hyperparameters (e.g.,

burn-in period, chain length, etc.) and, as we will show in Section 4, generating sufficient samples to reliably approximate the PPD may impose significant overhead. Klein et al. [2017] extended this parametric prior to also capture the effect of hyperparameter settings. In particular, they used a Bayesian neural network with a specialized learning curve layer, and trained this network using gradient-based MCMC on learning curve data from previously tested hyperparameter settings. While this approach is able to predict learning curves of previously unseen configurations, conditioning on the current partial learning curve requires retraining the Bayesian neural network online, which is costly. Recently, DyHPO [Wistuba et al., 2022] followed a similar dynamic HPO setup as Swersky et al. [2014], but used *deep* GPs [Damianou and Lawrence, 2013]. While deep GPs relax some of the standard GP assumptions, extrapolation abilities were not thoroughly analyzed, and DyHPO only predicts one epoch into the future. Finally, it is worth noting that, except for Domhan et al. [2015], all the aforementioned probabilistic approaches [Swersky et al., 2014, Klein et al., 2017, Chandrashekaran and Lane, 2017, Gargiani et al., 2019, Wistuba et al., 2022] utilize meta-learning across the learner's hyperparameter settings. While this is an interesting line of work, it limits applicability, and introduces confounding factors. We will therefore consider a simpler and more general setting in this work (see Section 3.1). Indeed, akin to the approach of Domhan et al. [2015], we operate without the assumption of access to data from previous runs employing different hyperparameter settings, nor do we assume the ability to generalize across these settings. Our results highlight that prior-data fitted networks (PFNs) offer a significantly more efficient and practical alternative to Markov Chain Monte Carlo (MCMC) methods. As categorized by Mohr and van Rijn [2022], Domhan et al. [2015] is the only comparable prior work within this category. This underlines the novelty and importance of our approach in the context of advancing current methodologies.

While we are the first to apply PFNs [Müller et al., 2022] to learning curve extrapolation, PFNs have previously been applied in different settings: Hollmann et al. [2023] used them to meta-learn a classifier for tabular data; Müller et al. [2023] as a surrogate model for Bayesian optimization; and most recently concurrent work by Dooley et al. [2023] as a zero-shot time series forecaster.

## 3 Methods

### 3.1 Bayesian learning curve extrapolation

Let $y_t \in [0, 1]$ represent the model performance (e.g., validation accuracy) at training step $t \in \{1, \ldots, m\}$. The problem we consider in this paper can be formulated as follows: Given a partial learning curve $y_1, \ldots, y_T$ up to some cutoff $T$, and a prior distribution $p(\boldsymbol{y})$ over learning curves, approximate the posterior predictive distribution (PPD) $q(y_{t'} \mid y_1, \ldots, y_T)$ for $T < t' \leq m$. We will further assume that we can calculate the relative probability density of $p(\boldsymbol{y})$, a requirement for MCMC, and that we can generate samples from $p(\boldsymbol{y})$, a requirement for PFNs. Figure 1 provides an illustration of Bayesian learning curve extrapolation, showcasing the posterior predictive distributions (PPDs) of the extrapolated curves generated by `LC-PFN` and MCMC, along with a few representative curves sampled from the prior distribution $p(\boldsymbol{y})$.

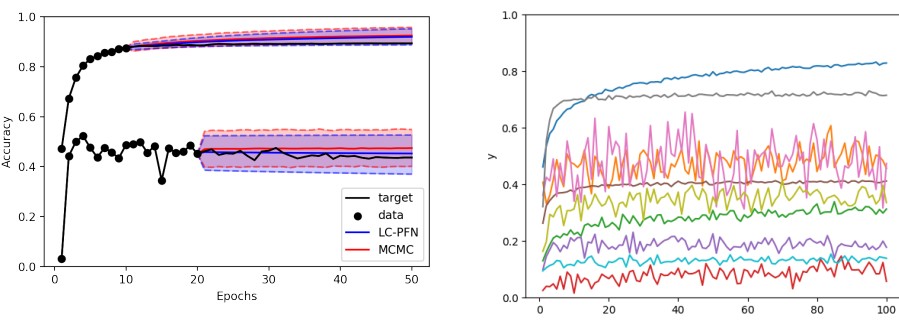

Figure 1: **(Left)** Visualization of Bayesian learning curve extrapolation. The plot shows the median and the 90% confidence interval of the PPDs inferred using MCMC and `LC-PFN`, given two partial empirical learning curves of 10 and 20 epochs, respectively, and the prior described in Section 3.2. **(Right)** Example of learning curves sampled from the prior $p(\boldsymbol{y})$.

While the fixed ranges for $y_t$ and $t$ are well-suited for modeling particular learning curves (e.g., accuracy over epochs), they also are restrictive. In Appendix A, we discuss the invertible normalization procedure we apply to support extrapolating, possibly diverging, iteration-based learning curves across a broad range of performance metrics (e.g., log loss).

## 3.2 Learning curve prior

Following Domhan et al. [2015], we model $y$ as a linear combination of $K$ basis growth curves $f_k$, each parameterized by $\boldsymbol{\theta}_k$, and i.i.d. additive Gaussian noise with variance $\sigma^2$, i.e.,

$$y_t \sim \mathcal{N}(f_{\text{comb}}(t|\boldsymbol{\xi}), \sigma^2) \quad \text{with} \quad f_{\text{comb}}(t|\boldsymbol{\xi}) = \sum_{k=1}^{K} w_k \cdot f_k(t|\boldsymbol{\theta}_k),$$

where we assume our model parameters

$$\boldsymbol{\xi} = (w_1, \ldots, w_K, \boldsymbol{\theta}_1, \ldots, \boldsymbol{\theta}_K, \sigma^2)$$

to be random variables with prior $p(\boldsymbol{\xi})$. Here, Domhan et al. [2015] assumed an uninformative prior (i.e., $p(\boldsymbol{\xi}) \propto 1$), with the exception of some hard constraints. We adopt a strictly more informative prior, because (i) the original prior puts almost all probability mass on parameterizations yielding invalid learning curves, e.g., $y_t \notin [0, 1]$; and (ii) we cannot practically sample from this prior having unbounded support, a requirement for PFNs.[1] Specifically, to mimic realistic learning curves we use bounded uniform weight priors $w_k \sim \mathcal{U}(0, 1)$, a low-noise prior $\log(\sigma^2) \sim \mathcal{N}(-8, 2)$, only allow curves with values in $[0, 1]$ and, like Domhan et al. [2015], only accept curves whose last value is higher than its first. Putting all of these together, our prior distribution thus takes the form:

$$p(\boldsymbol{\xi}) \propto \left( \prod_{k=1}^{K} p(w_k) \cdot p(\boldsymbol{\theta}_k) \right) \times p(\sigma^2) \times \mathbb{1}(f_{\text{comb}}(1|\boldsymbol{\xi}) < f_{\text{comb}}(m|\boldsymbol{\xi})) \times \left( \prod_{t=1}^{m} \mathbb{1}(f_{\text{comb}}(t|\boldsymbol{\xi}) \in [0, 1]) \right).$$

Finally, we limit ourselves to three parametric families of learning curves ($K = 3$, see Table 1).[2] These basis curves were chosen to capture a variety of growth trends and convergence behavior. We show examples of curves sampled from this prior in Figure 1 (right).

Table 1: Formulas of the three parametric basis curves and priors over their parameters.

| Reference name | Formula $f_k(t)$ | Prior $p(\boldsymbol{\theta}_k)$ |
|---|---|---|
| pow$_3$ | $c - at^{-\alpha}$ | $c \sim \mathcal{U}(0, 1.25) \quad a \sim \mathcal{U}(-0.6, 0.6) \quad \log(\alpha) \sim \mathcal{N}(0, 4)$ |
| Janoschek | $\alpha - (\alpha - \beta)e^{-\kappa t^\delta}$ | $\alpha \sim \mathcal{U}(0, 1) \quad \beta \sim \mathcal{U}(0, 2) \quad \log(\kappa) \sim \mathcal{N}(-2, 1) \quad \log(\delta) \sim \mathcal{N}(0, 0.25)$ |
| ilog$_2$ | $c - \frac{a}{\log(t+1)}$ | $c \sim \mathcal{U}(0, 1) \quad a \sim \mathcal{U}(-0.5, 0.5)$ |

## 3.3 Prior-data fitted networks (PFNs)

In this paper, we propose to use prior-data fitted networks (PFNs, Müller et al., 2022) instead of MCMC for learning curve extrapolation. PFNs are neural networks trained to perform approximate Bayesian prediction for supervised learning settings. That is, PFNs are trained to predict some output $y \in \mathbb{R}$, conditioned on an input $t$ and a training set $D_{train}$ of given input-output examples. The PFN is trained for this task with samples obtained from a prior over datasets $p(\mathcal{D})$. The loss function for training a PFN $q_\theta$ with parameters $\theta$ is the cross entropy $\ell_\theta = \mathbb{E}_{(t,y) \cup D_{train} \sim p(\mathcal{D})}[-\log q_\theta(y|t, D_{train})]$ for predicting the hold-out example's label $y$, given $t$ and $D_{train}$. Müller et al. [2022] proved that minimizing this loss over many sampled tasks $(t, y) \cup D_{train}$ directly coincides with minimizing the KL divergence between the PFN's predictions and the true PPD. In essence, the PFN meta-learns to perform approximate posterior inference on (meta-train) synthetic tasks sampled from the prior, and at inference time also does so for a (meta-test) real task.

---

[1]Note that the probability density of $p(y_t)$ is well-defined, a requirement for MCMC, but not for PFNs.

[2]We found that adding additional curves did not improve predictive performance significantly, and made the MCMC baseline even more expensive and less stable. Klein et al. [2017] also limited themselves to five basis curves for similar reasons.

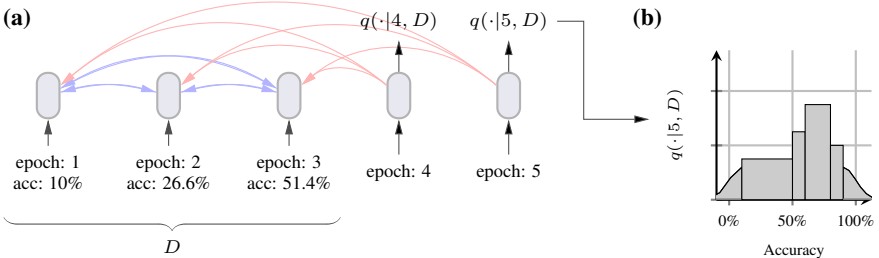

Figure 2: A visualization of our `LC-PFN` model on the task of predicting an accuracy over epochs curve. $D$ represents the epoch accuracies up to epoch 3 ($= T'$). Attention between test and training positions is shown using red and blue arrows, respectively. Plots based on Müller et al. [2022].

### 3.4 PFNs for Learning Curve Extrapolation (LC-PFNs)

To apply PFNs to learning curve extrapolation, we train them on learning curves sampled from a given prior over learning curves (in our case, the prior defined in Section 3.2). Specifically, the training set we condition on is the available partial learning curve up to some varying cutoff point $T'$, i.e., $D_{train} = \{(t', y_{t'})\}_{t'=1}^{T'}$, the input we condition on is the epoch $t \in \{T'+1, \ldots, m\}$ to predict for, and the desired output $y$ is the value of the learning curve at epoch $t$. During training, we randomly sample the cutoff points $T' \sim \mathcal{U}(0, m\text{ - }1)$ for every batch in order to learn to predict for initial learning curves of varying sizes. Figure 2 illustrates the information flow in a `LC-PFN` during learning curve extrapolation.

**LC-PFN architecture and hyperparameters** We use the PFN architecture proposed by Müller et al. [2022] and visualized in Figure 2 (a). That is, we use a sequence Transformer [Vaswani et al., 2017] and treat each pair $(t, y)$ (for train) and $t$ (for test) as a separate position/token. We encode these using a simple linear layer. We do not use positional encoding such that we are permutation invariant. Furthermore, the attention matrix is masked such that every position only attends to the training positions. This way training examples can attend to each other, but the test examples do not influence each other's predictions. Note that the output of the PFN with parameters $\theta$ is a distribution $q_\theta(y|t, D_{train})$. Following Müller et al. [2022], we discretize $q_\theta$ in a finite number of bins whose probability mass is predicted by the PFN, as is shown in Figure 2 (b). The size of each bin is set such that, under the prior, $y_t$ is equally likely to fall in each bin. The number of bins is a hyperparameter that we set to 1 000. The `LC-PFN` model further inherits hyperparameters from the Transformer, including the number of layers (`nlayers`), number of heads (`nheads`), embedding size (`emsize`), and hidden size (`nhidden`). We use four heads, a hidden size of 1 024, and conduct a thorough ablation study to investigate the effects of the number of layers and embedding size on the final performance, exploring a grid of values (see Table 2). We use a standard training procedure for all experiments, employing the Adam optimizer [Kingma and Ba, 2015] (learning rate 0.0001, batch size 100) with cosine annealing [Loshchilov and Hutter, 2017] with a linear warmup over the first 25% epochs of the training. Finally, we set $m = 100$, implying `LC-PFN` is trained for extrapolating sequences of up to 100 training steps (e.g., epochs). We found that most curves are shorter in practice, and when longer sequences are encountered, we subsample them as described in Appendix B.

## 4 Experiments

Our experiments aim to test the hypothesis that PFNs present a practical Bayesian approach to learning curve extrapolation. To this end, we first compare our `LC-PFN` approach against the MCMC approach of Domhan et al. [2015], using the same prior on samples generated from it (Section 4.1). Then, we extend the comparison to four real-world learning curve benchmarks (Section 4.2). Finally, we look beyond the quality of individual extrapolations and evaluate the potential of `LC-PFN` in the context of predictive early stopping to accelerate model selection (Section 4.3).

## 4.1 Extrapolating samples of the prior

The goal of this first experiment is to assess the ability of `LC-PFN`s and `MCMC` to approximate the true posterior predictive distribution (PPD). To avoid artifacts due to out-of-distribution data, in this experiment, we use curves sampled from the prior (defined in Section 3.2). Furthermore, we modified the original implementation of MCMC [Domhan et al., 2015], to use the curve prior we proposed in Section 3.2. In the following, we refer to this MCMC variant as `MCMC-PP` and denote the one using the original prior [Domhan et al., 2015] as `MCMC-OP` (used in Section 4.2). Since the `LC-PFN` and `MCMC-PP` methods are both (approximate) Bayesian inference methods, using the same prior, they aim to approximate the same true target PPD, given a partial learning curve.

As a performance metric, we evaluate the log-likelihood (LL) of the unseen data (right-censored curve) under the inferred PPD. We use this metric, also known as *logarithmic score*, to assess a model's ability to infer the remaining part of the curve based on the initial observed values. A benefit of this metric is that it measures the quality of the PPD as a whole (rather than merely focusing on the error associated to a specific PPD statistic) and, assuming data is generated by the prior, the exact PPD maximizes this metric.

Importantly, we vary the cutoff, i.e., the percentage of the observed curve used as input, to better assess the model's performance across different amounts of available information. Furthermore, to allow a more comprehensive comparison, we vary the hyperparameters of both `LC-PFN` and `MCMC-PP`. For `LC-PFN`, we vary the embedding size (`emsize`), the number of layers (`nlayers`), and the total number of learning curves used during training (`nb_data`). For `MCMC-PP`, we vary the number of chains generated by the `emcee` [Foreman-Mackey et al., 2013] ensemble sampler (`nwalkers`), the length of each chain

Table 2: Grid of hyperparameter values evaluated for `MCMC-PP` and `LC-PFN`.

| | Hyperparameters |
|---|---|
| MCMC-PP | $\text{nsamples} \in [100, 250, 500, 1000, 2000, 4000]$ $\text{nwalkers} \in [26, 50, 100]$ $\text{burn-in} \in [0, 50, 100, 500]$ $\text{thin} \in [1, 10, 100]$ |
| LC-PFN | $\text{nb\_data} \in [100k, 1M, 10M]$ $\text{emsize} \in [128, 256, 512]$ $\text{nlayers} \in [3, 6, 12]$ |

(`burn-in` + `nsamples`), the part of the chain omitted to account for mixing (`burn-in`), and the sub-sample frequency (`thin`). The considered values for each hyperparameter are summarized in Table 2.

We conducted the comparison on 10 000 sampled curves. Figure 1 shows a few curve examples, as well as inferences using `LC-PFN` and `MCMC-PP` given the data of the first 10 - 20 epochs (cutoff). We observe that both predicted median and uncertainties are indeed similar. More inference examples, with different cutoffs can be found in Appendix C.4.

**Results**  Figure 3 displays the average log-likelihood across `MCMC-PP` / `LC-PFN` inferences for varying hyperparameters and a 10% cutoff. The log-likelihood is shown w.r.t. runtime, which is measured as the average wall-clock time for a single inference on a single Intel(R) Xeon(R) Gold 6242 CPU. Note that this inference time includes both the fit and prediction times for MCMC variants. Table 3 provides results on higher cutoffs (20%, 40%, 80%) and corresponding runtimes for three variants of each method (M1-3, P1-3), labeled in Figure 3. Generally, the `LC-PFN` variants (left side of figure 3) are significantly faster than `MCMC-PP` variants (right side). `LC-PFN` always ran in less than 0.1 seconds while the best `MCMC-PP` (M3) took over 100 seconds. Figure 3 also offers insights into the importance of `LC-PFN` and `MCMC-PP` hyperparameters. For `MCMC-PP`, both the cost and quality of inference increase with longer chain lengths and higher cutoffs. For `LC-PFN`, the inference cost increases with the model complexity which is closely related to the number of trainable parameters. Inference quality positively correlates with model size ("larger is better"), and the number of data `LC-PFN` was trained on. Among the hyperparameter grid we examined (Table 2), except for the smallest model (P1), all `LC-PFN` variants that were trained on 10M samples produce higher log-likelihood than the best `MCMC-PP` variant (M3). In particular, an `LC-PFN` (P2) with 3 layers, embedding size 256, trained on 10M samples achieved better performance (log-likelihood of PPD) than the best `MCMC-PP`, but more than 15 000 times faster. We also find that while the runtime of the best `MCMC-PP` can be reduced (with minor loss of quality) by using thinning (M2), the better `LC-PFN` is still approximately 7 000 times faster. Finally, it is important to note that training the largest `LC-PFN` (P3, 10M samples with 26M parameters) on the prior took approximately eight hours (single CPU, single RTX2080 GPU), but this cost is incurred only *once* for all of our experiments.

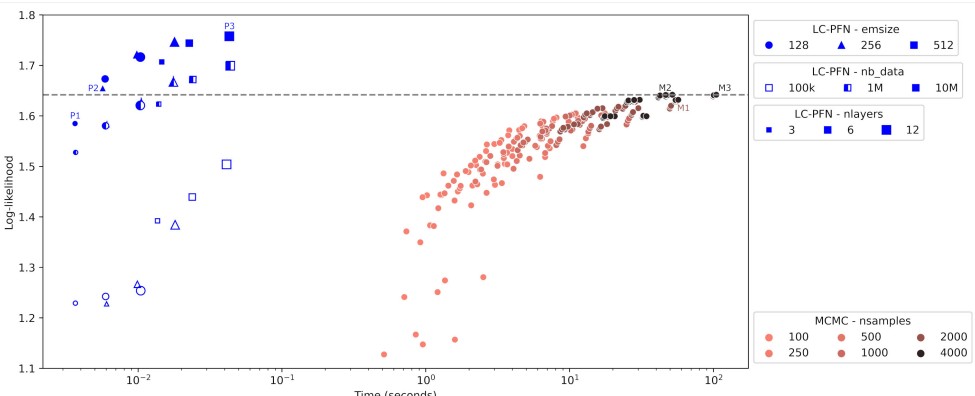

Figure 3: Runtime (lower is better) *vs* log-likelihood of the true curve under the PPD (higher is better), with 10% of the curve observed. See Figure 9 in Appendix C.1 for higher cutoffs. Blue and red markers correspond respectively to `LC-PFN` and `MCMC-PP` with varying hyperparameters values. The `M1-3`/`P1-3` labels refer to the PFN/MCMC variants listed in Table 3. The horizontal dashed line indicates the performance of the best MCMC variant.

Table 3: Comparison of three `LC-PFN` and `MCMC-PP` variants on prior curves in terms of log-likelihood (higher is better) at 10%, 20%, 40%, and 80% cutoffs. Here, `M1` corresponds to the configuration used in Domhan et al. [2015]. Please refer to Table 5 for more comprehensive results.

| Label | Method | Parameters | 10% | 20% | 40% | 80% | Avg. Runtime (s) |
|-------|--------|-----------|-----|-----|-----|-----|------------------|
| M1 | MCMC | nsamples=2000, nwalkers=100, burn-in=500, thin=1 | 1.628 | 1.939 | 2.265 | 2.469 | 54.401 |
| M2 | MCMC | nsamples=4000, nwalkers=100, burn-in=100, thin=100 | 1.641 | 1.958 | 2.277 | 2.477 | 45.160 |
| M3 | MCMC | nsamples=4000, nwalkers=100, burn-in=500, thin=1 | 1.642 | 1.956 | 2.285 | 2.486 | 103.151 |
| P1 | PFN | nb_data=10M, nlayers=3, emsize=128 | 1.58 | 1.99 | 2.28 | 2.43 | 0.004 |
| P2 | PFN | nb_data=10M, nlayers=3, emsize=256 | 1.65 | 2.04 | 2.35 | 2.49 | 0.006 |
| P3 | PFN | nb_data=10M, nlayers=12, emsize=512 | **1.76** | **2.13** | **2.40** | **2.52** | 0.050 |

## 4.2 Extrapolating real-world learning curves

While evaluation on data from the prior gives us a controlled setting to analyse quality and cost of the PPD approximation, performance on real-world learning curves is essential for practical usefulness. This second experiment aims to extend the previous comparison of MCMC and `LC-PFN` to real-world learning curve benchmarks.

We consider the best-performing variants of `LC-PFN` and `MCMC-PP` according to the average log-likelihood they obtained in the first experiment. For `LC-PFN`, the optimal variant (`P3`) features an embedding size of 512 and 12 layers, resulting in a total of 26M trainable parameters, and is trained on 10 million prior curves. For `MCMC-PP`, the optimal configuration (`M3`) involves a chain length of 4 500, 100 walkers, 500 burn-in samples, without thinning. As an additional baseline, we include `MCMC-OP`, the original MCMC variant proposed by Domhan et al. [2015], which uses the original hyperparameters and curve prior (11 basis curves and uninformative prior over the curve parameters).

**Benchmarks** To evaluate the generalization capabilities of our model, we consider a diverse set of real-world curves. Our dataset comprises 20 000 learning curves, sourced from four distinct benchmarks: `LCBench` [Zimmer et al., 2021], `NAS-Bench-201` [Dong and Yang, 2020], `Taskset` [Metz et al., 2020] and `PD1` [Wang et al., 2022], each contributing 5 000 curves, randomly selected from specific subtasks. These benchmarks and subtasks were chosen to span a broad spectrum of supervised deep learning problems, training MLP (`LCBench`), CNN (`NAS-Bench-201`), RNN (`Taskset`), and Transformer (`PD1`) architectures on input modalities ranging from tabular data (`LCBench`), text (`Taskset` and `PD1`), protein sequence (`PD1`), to vision problems (`NAS-Bench-201`). From `LCBench` and `NAS-Bench-201` we use validation accuracy curves whereas from `Taskset` and `PD1` the log loss validation curves. In terms of curve length, `LCBench` and `Taskset` cover 50 epochs, `NAS-Bench-201` contains up to 200 epochs, and `PD1` curves have varying lengths (22 - 1414). Further details, including sample curves, on these benchmarks are provided in Appendix B.

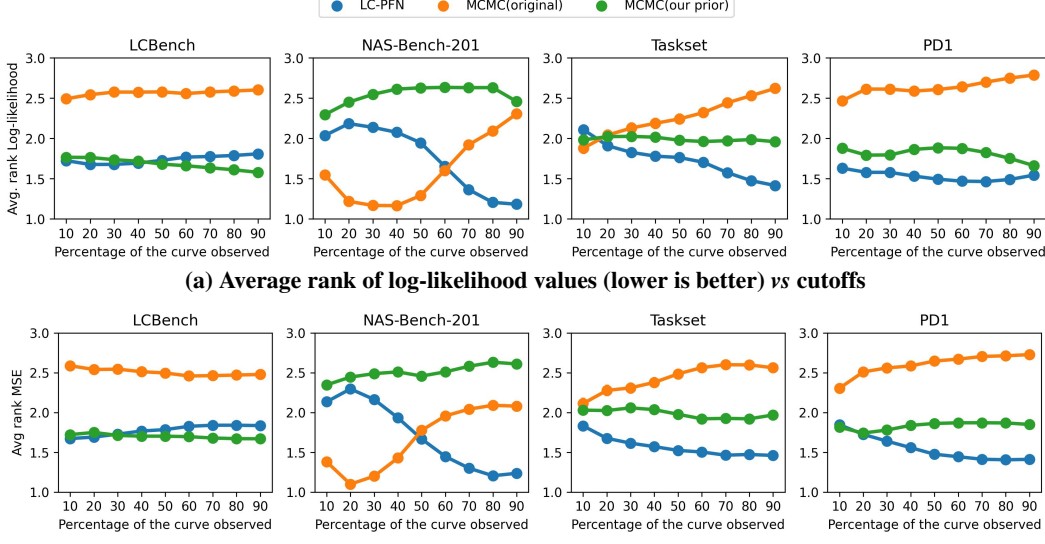

**(a) Average rank of log-likelihood values (lower is better) *vs* cutoffs**

**(b) Average rank of mean squared error (MSE) values (lower is better) *vs* cutoffs**

Figure 4: Comparison of `LC-PFN` with two MCMC variants on three real-data benchmarks.

**Metrics** Our focus lies on the relative performance of MCMC and `LC-PFN`, as absolute performance is significantly influenced by the choice of prior. We consider the log-likelihood and the mean squared error (MSE) of the predictions as metrics. Here, the MSE is calculated w.r.t. the median of the PPD.

For each benchmark, we report the average rank of these metrics to aggregate results from different curves, as supposed to the average values. While the latter better captures performance differences, it is very sensitive to outliers and scale-dependent. When computed in normalized space, it would strongly depend on our choice of normalization parameters (see Appendix A).

**Results** For each benchmark, Figures 4a and 4b display the average rank obtained by each method in terms of log-likelihood and MSE, respectively, where ranks are averaged across all 5 000 curves. We do not include error bars as the standard errors are neglectable (less than 0.02). In summary, we observe similar trends for both metrics on all benchmarks. `LC-PFN` is never truly outperformed by `MCMC-PP`. On `LCBench` both methods rank similarly, with `LC-PFN` being slightly worse at high cutoffs. `LC-PFN` ranks better on `PD1`, `NAS-Bench-201`, and `Taskset`. `MCMC-OP` performs clearly worse on `LCBench`, `Taskset`, and `PD1`. On `NAS-Bench-201`, `MCMC-OP` performs best for lower cutoffs, outperforming `MCMC-PP`, suggesting that `NAS-Bench-201` curves are better captured by the original prior. Figure 11 and Figure 12 in Appendix C.2 show the log-likelihoods and MSEs, respectively, for all three methods, for each curve and cutoff, per benchmark, providing a more detailed perspective.

### 4.3 Application: Extrapolation-based early stopping in model selection

Thus far, we have shown that `LC-PFN` produces extrapolations of similar or better quality to `MCMC`, at a small fraction of the cost. However, these extrapolations are not perfect. In many cases, the practical relevance of any errors can only be assessed in the context of a specific application.

In this final experiment, we thus consider a model selection setting where after every epoch of training we have the choice between (i) continuing the current training, or (ii) stopping early ($T < m$) and starting a new training run using a different training pipeline (e.g., model architecture, hyperparameters, etc.). Here, we assume that runs cannot be resumed once stopped and that the order in which training pipelines are to be considered is given. Our objective is to obtain high-quality models as quickly as possible, by stopping suboptimal training runs as early as possible. This setting is also known as vertical model selection [Mohr and van Rijn, 2022] and was also considered by Domhan et al. [2015].

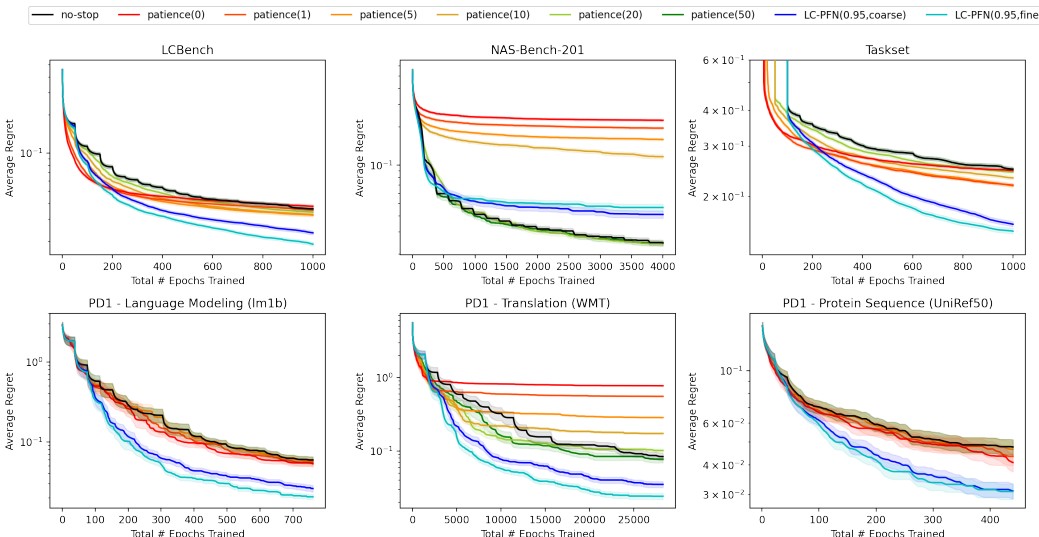

Figure 5: Comparison of our `LC-PFN` based early stopping mechanism to naive baselines `no-stop` (no stopping, train for the full budget $m$) and `Patience(k)` (stop training after `k` epochs without improvement) for vertical model selection where runs are considered in a fixed order and cannot be resumed. Shown is the anytime regret (lower is better) different approaches achieve after a total number of training epochs, averaged per benchmark and across 40 orderings per task.

We consider the extrapolation-based termination criterion proposed by Domhan et al. [2015], but use `LC-PFN` instead of `MCMC` to predict the likelihood $\Pr(y_{t'} > y_{\text{best}} \mid y_1, \ldots, y_T)$ that the current run will at epoch $t'$ obtain a model better than the best obtained by any run thus far ($y_{\text{best}}$), for $T < t' \leq m$ and decide to stop the current run if that probability does not exceed some fixed threshold $\delta$ (at any point). In our experiments, we use confidence level $1 - \delta = 0.95$ as Domhan et al. [2015]. Note that this criterion can be applied at every step or only at specific cutoffs. To simulate the effect of varying granularity, we consider a coarse-grained variant with 4 cutoffs $T \in \{\lceil 0.1m \rceil, \lceil 0.2m \rceil, \lceil 0.4m \rceil, \lceil 0.8m \rceil\}$, and a fine-grained variant with $T \in \{T \mid 1 < T < m\}$. We investigate alternative choices for cutoffs and confidence levels in Appendix C.3.2.

Following Domhan et al. [2015], we compare against a black box approach `no-stop` that does not implement early stopping. We further compare against a criterion `Patience(k)` that implements the popular heuristic to terminate a training run when model performance did not improve for $k$ epochs. For evaluation, we consider the same benchmarks as in Section 4.2 (for details, see Appendix B). We chose the total budget for model selection to correspond to 20 full runs ($20m$). For each task, we consider the training runs in 40 different random orderings. This totals 2 120 model selection experiments per method, spread across the 53 different tasks.

**Results** Figure 5 shows the anytime performance of all methods in our comparison, on each of the benchmarks, in terms of regret. Here, regret is the absolute difference in performance between the best model obtained thus far and the best model attained by any run on the task. Results are averaged over the different run orderings. For `LCBench`, `NAS-Bench-201`, and `Taskset` results are further averaged across all tasks (results for individual tasks can be found in Appendix C.3.1). The shaded area corresponds to $\pm 1$ standard error. On all benchmarks, except for `NAS-Bench-201`, we observe that `LC-PFN` based termination criteria clearly perform best. Also, the fine-grained variant performs better on average, suggesting that errors in inference are compensated for by the time saved by stopping runs earlier. In terms of expected speed-up, this `LC-PFN` variant obtains an expected regret lower than that obtained by `no-stop`, approximately 3.3× faster on `LCBench` and `Taskset`. Looking at individual tasks, we note 2-6× speed-ups on all 3 `PD1` tasks, all 12 `Taskset` tasks, and 30 of the 35 `LCBench` tasks (we obtain 1-2 × speed-ups on the remaining 5). On `NAS-Bench-201`, we find that all termination criteria considered, including standard Patience heuristics, fail on all 3 tasks; this is likely related to the particular shape of learning curves on this benchmark, having an inflection point (see Figure 8 and Figure 19), not resembling any in the prior.

Finally, in terms of overhead, the computational costs of the `LC-PFN` inferences per model selection experiment range from 3 seconds (coarse-grained) up to 1 minute (fine-grained), and are negligible compared to the cost of the 20 full training runs of deep neural networks.

# 5 Summary, limitations, and future research

We presented the first work using prior-data fitted networks (PFNs) for Bayesian learning curve extrapolation. We show that our `LC-PFN` obtains qualitatively similar extrapolations, for a wide variety of learning curves, more than $10\,000\times$ faster than the MCMC method proposed by Domhan et al. [2015]. These inferences are now *fast enough* (under 100 milliseconds on CPU, and even less on GPU), to be used in the context of online learning, at virtually no overhead. This opens up a wide variety of possible applications, e.g., to speed up automated model selection in AutoML and HPO by discarding poor configurations early. It would be interesting to integrate `LC-PFN` as a new termination criterion in existing deep learning libraries.

Often, we also have more data available than a single partial learning curve, e.g., other curves on the same task, their hyperparameters, and/or curves of the same method on a different task, and meta-features. Previous work [Swersky et al., 2014, Klein et al., 2017, Wistuba et al., 2022, Ruhkopf et al., 2022] has already exploited this, and we could explore the potential of using PFNs for few-shot *in-context* meta-learning, by feeding the model multiple curves and hyperparameters as input.

While for a fair comparison to Domhan et al. [2015], reusing the original code, our prior (Section 3.2) closely resembled the one of Domhan et al. [2015], future work could improve upon this prior and overcome its limitations (e.g., on the `NAS-Bench-201` tasks) by modelling divergence, slow start, double-descent, correlated heteroscedastic noise, etc.

Finally, PFNs, unlike other Bayesian methods, must learn the prior from data, which implies that the prior must be generative. Also, it suggests that high entropy priors may present challenges. Future research should investigate these limitations and how to overcome them.

# 6 Acknowledgments and disclosure of funding

We acknowledge funding by the European Union (via ERC Consolidator Grant Deep Learning 2.0, grant no. 101045765), TAILOR, a project funded by EU Horizon 2020 research and innovation programme under GA No 952215, the state of Baden-Württemberg through bwHPC and the German Research Foundation (DFG) through grant numbers INST 39/963-1 FUGG and 417962828. Views and opinions expressed are however those of the author(s) only and do not necessarily reflect those of the European Union or the European Research Council. Neither the European Union nor the granting authority can be held responsible for them.

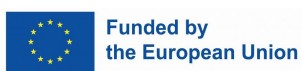

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

# A    Learning curve normalization

Not all learning curves of interest may resemble those of the prior we described in Section 3.2. For example, when minimizing the negative log-likelihood loss (log loss), learning curves will mostly decrease (vs. increase), be convex (vs. concave), and could take values that exceed 1.0 (there is no hard upper bound for log loss).

One approach would be to define a prior and train a specialized PFN for every performance measure. In this work, we use a more general approach: We apply a normalization procedure that allows us to accurately extrapolate a wide variety of learning curves using a single PFN, without retraining or fine-tuning. We consistently apply this procedure to all inferences involving real learning curves (i.e., the experiments described in Section 4.2 and Section 4.3), even if observations are naturally constrained to $[0, 1]$, e.g., accuracy curves.

**Step 1: Normalize the partial learning curve:**    Let $y_i^o$ be the $i^{\text{th}}$ performance observation. We start by normalizing it using a generalized logistic sigmoid transform $y_i = g_{\boldsymbol{\lambda}}(y_i^o)$, that is re-parametrized by the following five parameters ($\boldsymbol{\lambda}$):

**min?**  A Boolean specifying that we expect learning to minimize the performance measure. E.g., this would be true for error rate and log loss, and false for accuracy.

$\mathbf{l_{hard}, u_{hard}}$  These are possibly infinite hard lower / upper bounds for model performance. This would, e.g., be 0 / 1 for accuracy and error rate; and $0 / +\infty$ for log loss.

$\mathbf{l_{soft}, u_{soft}}$  These are finite soft lower/upper bounds for model performance. They specify the range in which we expect performance values (that we care to distinguish between) to lie in practice. For accuracy and error rate, one could set these equal to the hard bounds, whereas for log loss one could, e.g., use an estimate of the loss for the untrained network (i.e., at epoch 0) as $u_{soft}$ and $l_{soft} = l_{hard}$, or if available, choose $l_{soft}$ to be an optimistic estimate of performance at convergence (e.g., state-of-the-art). Narrower ranges will result in more accurate inferences.

Specifically, we perform a linear transformation, followed by a logistic sigmoid, another linear transform, and a minimization conditional reflection around $\frac{1}{2}$, i.e.,

$$g_{\boldsymbol{\lambda}}(y_i^o) = \mathrm{cr}_{0.5}\left(\frac{c}{1 + e^{-a(x-b)}} + d\right)$$

where

$$\mathrm{cr}_{0.5}(y) = \begin{cases} 1 - y & \text{if min?} \\ y & \text{if } \neg\text{min?} \end{cases} \qquad a = \frac{2}{u_{soft} - l_{soft}} \qquad b = -\frac{u_{soft} + l_{soft}}{u_{soft} - l_{soft}}$$

$$c = \frac{1 + e^{-a(u_{hard}-b)} + e^{-a(l_{hard}-b)} + e^{-a(u_{hard}+l_{hard}-2b)}}{e^{-a(l_{hard}-b)} - e^{-a(u_{hard}-b)}} \qquad d = \frac{-c}{1 + e^{-a(l_{hard}-b)}}$$

Note that for $\boldsymbol{\lambda} = (\text{False, } \text{-}\infty, -1, 1, \infty)$ this reduces to the canonical logistic sigmoid $y_i = \frac{1}{1+e^{-y_i^o}}$. This transform will be approximately linear (shape preserving) for $y_i^o \in [l_{soft}, u_{soft}]$, and the tails of the sigmoid will gradually squish values outside this range to $[0, 1]$, supporting unbounded performance measures. Figure 6 visualizes this projection in general, and provides examples for accuracy with $\boldsymbol{\lambda} = (\text{False}, 0, 0, 1, 1)$ and log loss with $\boldsymbol{\lambda} = (\text{True}, 0, 0, \log(10), \infty)$.

**Step 2: Infer using normalized data:**    Next, we perform Bayesian learning curve extrapolation on the normalized partial curve $\boldsymbol{y}$, with our usual prior $p(\boldsymbol{y})$, resulting in an approximation of the PPD in the transformed space.

**Step 3: Inverse transform the PPD:**    Finally, we can obtain the property of interest of the PPD in the original space by applying the inverse transform, given by

$$g_{\boldsymbol{\lambda}}^-(y_i^p) = \frac{\log\left(\frac{\mathrm{cr}_{0.5}(y_i^p)-d}{c-(\mathrm{cr}_{0.5}(y_i^p)-d)}\right) - b}{a}$$

We use order-based statistics (e.g., median or other percentiles) in our experiments, to which we can simply apply $g_{\boldsymbol{\lambda}}^-$ directly since it is a monotonic transform. For other statistics (e.g., mean, variance) we may need to resort to Monte Carlo estimation, applying $g_{\boldsymbol{\lambda}}^-$ to samples of the PPD.

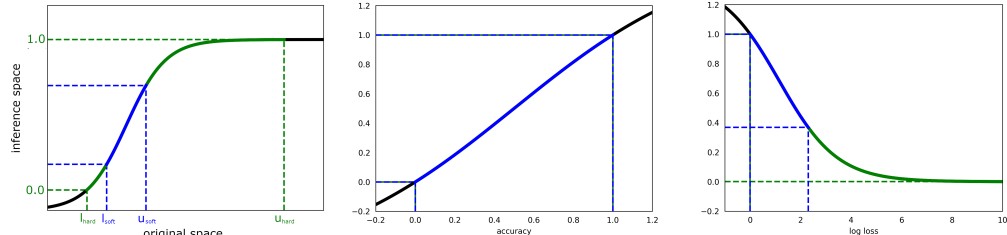

Figure 6: **left:** The generic logistic sigmoid transformation described in Appendix A. and used to normalize learning curves to support a wide range of possibly unbounded performance metrics. **middle/right:** An example of the normalization for curves maximizing accuracy / minimizing log loss, using the transformation.

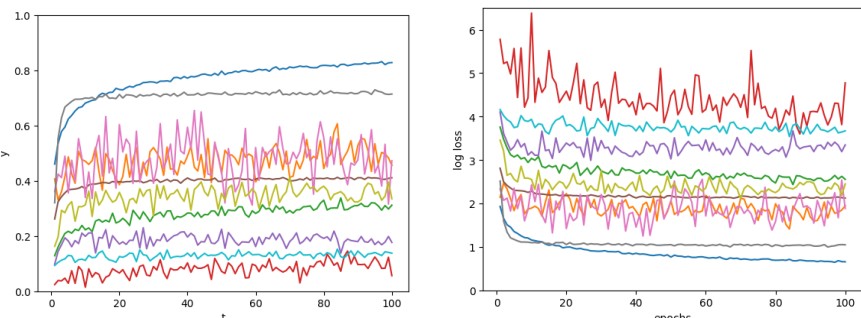

Figure 7: Sample of 10 curves taken i.i.d. from the prior (Section 3.2). **(Right)** The same sample after applying the inverse transformation $g_{\boldsymbol{\lambda}}^-$ to obtain samples from the log loss prior as described in Appendix A, using the transform shown in Figure 6 (right).

By applying a given normalization, we effectively transform our prior, i.e., $p(\boldsymbol{y}^o) = g_{\boldsymbol{\lambda}}^-(p(\boldsymbol{y}))$. To get some intuition of what that prior looks like, we can sample from $p(\boldsymbol{y}^o)$ by applying $g_{\boldsymbol{\lambda}}^-$ to samples of our prior $p(\boldsymbol{y})$, which is useful for fine-tuning the parameters of this transformation. Figure 7 (right) shows samples of the log loss prior for $\boldsymbol{\lambda} = (\text{True}, 0, 0, \log(10), \infty)$.

## B  Detailed benchmark description

To evaluate the generalization capabilities of `LC-PFN`, we consider a diverse set of real-world curves. The reduce computational cost and improve reproducibility, we source these from four existing benchmarks that provide a collection of learning curves for a wide variety of supervised learning tasks. In what follows, we describe each of these benchmarks in more detail, the subset of tasks and curves we considered, and any preprocessing we did. Table 4 provides an overview of the characteristics of each benchmark and Figure 8 (a) visualizes the curves selected from each benchmark.

**LCBench**   provides learning curve data for 2 000 configurations of AutoPytorch [Zimmer et al., 2021] trained for 50 epochs on 35 tabular classification datasets from the AutoML benchmark [Gijsbers et al., 2019]. All of these configurations use momentum SGD to train an MLP, but with varying number of layers and units per layer, and varying optimization hyperparameters (batch size, learning rate, momentum, L2 regularization, dropout rate). In Section 4.2, we consider a subset of 5 000 curves selected uniformly at random from all 70 000 validation accuracy curves in the benchmark. In Section 4.3, we consider all 2 000 validation accuracy curves, for every task, in 40 different orderings.

**NAS-Bench-201**   is a benchmark for Neural Architecture Search methods [Dong and Yang, 2020]. It provides learning curve data for training 15 625 different architectures for 200 epochs on three different image classification datasets (CIFAR-10, CIFAR-100, and ImageNet16-120) for three different random seeds. Other aspects of the training pipeline (e.g., optimizer, hyperparameters)

Table 4: Overview of the characteristics of the real-world learning curve benchmarks we used in Section 4.2 and Section 4.3. For each benchmark, the last columns list the normalization parameters used (see Appendix A), where $y_0$ corresponds to the performance of the untrained model.

| benchmark | # subtasks | model architecture | input modality (datasets) | length (# epochs) | metric (split) | min? | $l_{hard}$ | $l_{soft}$ | $u_{soft}$ | $u_{hard}$ |
|---|---|---|---|---|---|---|---|---|---|---|
| `LCBench` | 35 | MLP | Tabular Classification (OpenML) | 50 | accuracy (val) | False | 0 | 0 | 1 | 1 |
| `NAS-Bench-201` | 3 | CNN | Image Classification (Cifar-10, Cifar-100, ImageNet16-120) | 200 | error rate (val) | True | 0 | 0 | 1 | 1 |
| `Taskset` | 12 | RNN | Text Classification (Sentiment Analysis - IMDB) | 50 | log loss (val) | True | 0 | 0 | $y_0$ | $+\infty$ |
| `PD1` | 3 | Transformer | Text (Language Modeling - lm1b) | 38 | log loss (val) | True | 0 | 3.46 | $y_0$ | $+\infty$ |
| | | Transformer | Text (Translation - WMT) | 1414 | log loss (val) | True | 0 | 1.68 | $y_0$ | $+\infty$ |
| | | Transformer | Protein sequences (UniRef50) | 22 | log loss (val) | True | 0 | 2.60 | $y_0$ | $+\infty$ |

are fixed. We use the validation error rate curves provided through the Syne Tune [Salinas et al., 2022] interface to this benchmark. As discussed in Section 3.4, the `LC-PFN` we consider is trained to extrapolate curves up to length 100 ($m$).[3] To handle the 200 epoch curves from `NAS-Bench-201`, we chose to subsample them, feeding only every second observation into the `LC-PFN`. In Section 4.2, we consider a subset of 5 000 curves selected uniformly at random from all 140 625 error rate curves in the benchmark. In Section 4.3, we consider all 46 875 error rate curves, for each of the three datasets, in 40 different orderings.

**Taskset** [Metz et al., 2020] provides a total of roughly 29 million learning curves for over 1 162 different deep learning tasks. Here, a task is defined as optimizing a given neural network architecture for a given supervised learning problem, and the curves correspond to using 5 different optimizers, using 1 000 different hyperparameter settings, and 5 different random seeds. Following Wistuba et al. [2022], we only use the validation log loss curves of a small subset of 12 tasks, that consider training different RNN architectures, with varying architectural parameters, for 50 epochs on the IMDB sentiment analysis dataset [Maas et al., 2011], a binary text classification task. As our objective was to evaluate the robustness of our approach, we avoided excluding ill-behaved (e.g., diverging) curves. However, upon inspecting the data, we found that on some of these tasks up to 90% of the curves fail to significantly improve upon the initial model performance, some of which diverge almost instantly. In combination with the normalization procedure described in Appendix A, the majority of curves are effectively mapped onto the same narrow range, producing a constant trend (at 0 in case of divergence). While this is reasonable in practice, and both `LC-PFN` and `MCMC` variants predict this constant trend with very high confidence, it does create a bias in our evaluation towards methods excelling at extrapolating poor curves. To eliminate and investigate this bias, we selected the 5 000 curves used in Section 4.2, such that 2 500 are taken from the top 10% best[4] curves per task, and 2 500 from the 90% others. In Figure 4, we compared methods on the "good" curves only, an evaluation on the 2 500 "bad" curves is presented in Figure 8 (b). In Section 4.3, we do not make this distinction and consider all 25 000 curves per task, in 40 different random orderings.

**PD1** is a recent benchmark that Wang et al. [2022] describe as collecting "a large multi-task hyperparameter tuning dataset by training tens of thousands of configurations of near-state-of-the-art models on popular image and text datasets, as well as a protein sequence dataset". We access this benchmark through the synetune [Salinas et al., 2022] library interface, which provides learning curve data for 23 tasks, where the number of curves provided, as well as the curve length, varies per task. Here, we limit our selection to log loss curves for the three tasks that consider training Transformer architectures for 22 epochs on language modeling (lm1b), 1414 epochs on translation (WMT), and 38 epochs on protein sequences (UniRef50). To handle the very long learning curves on the translation task, we subsample these aggressively, feeding only every 14th observation into the `LC-PFN`. In Section 4.2, we select 5 000 curves from these three tasks uniformly at random. In Section 4.3, we consider all curves per task, in 40 different random orderings. Finally, we use the optimal loss achieved by any of the runs as $l_{soft} > 0$ in our normalization. While unknown in practice, we do not expect qualitative differences in evaluation if a rough estimate were to be used instead.

---

[3]This was our first choice of $m$ and we retained it throughout the project. We do not anticipate serious issues when scaling $m$ up to 1 000, except for increased training times.

[4]Curves are ranked based on the lowest validation loss obtained at any point during training.

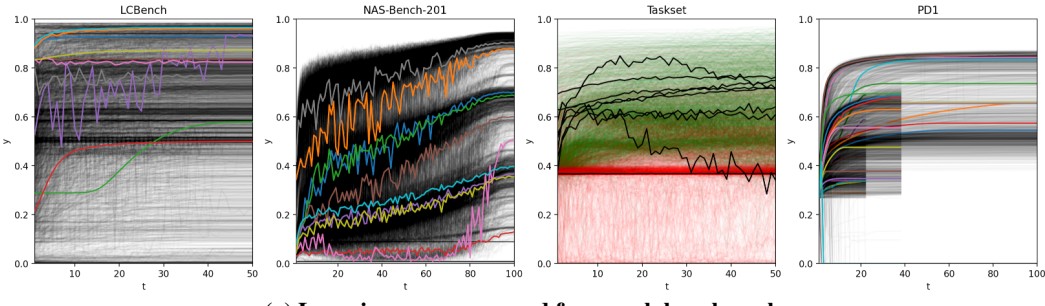

**(a) Learning curves sourced from each benchmark**

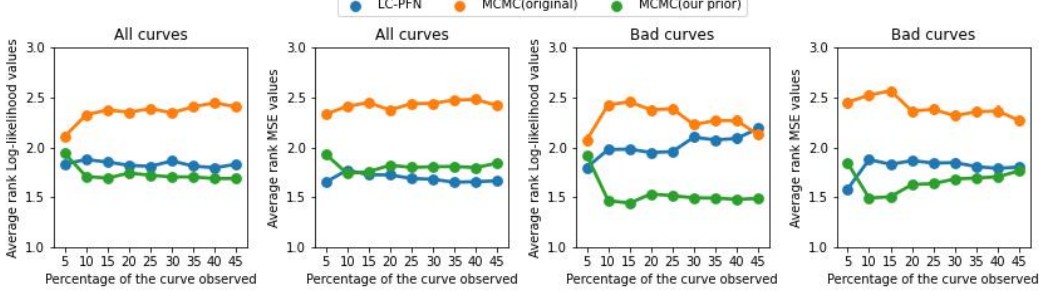

**(b) Quality of curves vs. quality of extrapolations on `Taskset` (average rank, lower is better)**

Figure 8: (a) Illustration showing the 5 000 curves used for each of the four benchmarks in our experiments in Section 4.2. All curves are shown after normalization and subsampling, and a random subset of individual curves is highlighted. For `Taskset`, "good" / "bad" curves (top 10% / bottom 90% in their task) are shown in green / red, respectively. (b) Average rank of Log-likelihood and MSE values on Taskset curves for the three methods (`LC-PFN`, `MCMC-PP`, and `MCMC-OP`). The two leftmost plots show results on the 5 000 sampled `Taskset` curves. The two rightmost plots provide the same comparison, but specifically for the 2500 "bad" curves. Here, we observe that `MCMC-PP` performs better on the "bad" curves, which we believe can be attributed to the discretization of the PPD in 1000 bins, limiting the maximal confidence / accuracy of `LC-PFN`, since the majority of these curves are quasi-constant (fall in the same bin) after normalization.

## C    Additional analyses

### C.1    Extrapolating samples of the prior

#### C.1.1    Detailed results of section 4.1

Figure 9 provides a visual representation of the log-likelihood values on prior curves for higher cutoffs, similar to Figure 3. As expected, the difference in log-likelihood values between `LC-PFN` and `MCMC-PP` decreases as more points of the curve are observed. The detailed results can be found in Table 5. Both Figure 9 and Table 5 clearly demonstrate that certain variants of `LC-PFN`, particularly those trained with 10M examples, consistently outperform the best variant of `MCMC-PP` across all considered cutoffs.

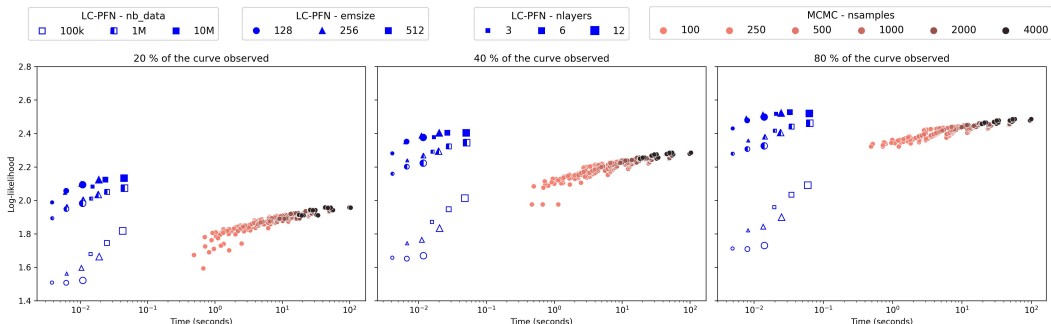

Figure 9: Comparison of `LC-PFN` and `MCMC-PP` with varying hyperparameters values on prior curves in terms of average runtime (lower is better) and average log-likelihood (higher is better) for different cutoff values (20%, 40%, 80%)

Table 5: Comparison of the 25 best `LC-PFN` and `MCMC-PP` variants on prior curves in terms of log-likelihood (higher is better) at 10%, 20%, 40%, and 80% cutoffs. Values in brackets correspond to one standard error.

| Method | Parameters | 10% | 20% | 40% | 80% | Avg. Runtime (s) |
|---|---|---|---|---|---|---|
| MCMC-PP | nsamples=2000, nwalkers=100, burn-in=500, thin=10 | 1.628 (0.01) | 1.939 (0.011) | 2.265 (0.01) | 2.469 (0.009) | 29.944 (7.6E-01) |
| MCMC-PP | nsamples=2000, nwalkers=100, burn-in=500, thin=1 | 1.628 (0.01) | 1.939 (0.011) | 2.265 (0.01) | 2.469 (0.009) | 54.401 (3.1E-01) |
| MCMC-PP | nsamples=4000, nwalkers=50, burn-in=0, thin=10 | 1.629 (0.01) | 1.942 (0.011) | 2.266 (0.009) | 2.467 (0.009) | 28.783 (6.2E-01) |
| MCMC-PP | nsamples=4000, nwalkers=50, burn-in=0, thin=1 | 1.629 (0.01) | 1.943 (0.011) | 2.266 (0.009) | 2.467 (0.009) | 53.292 (4.3E-01) |
| MCMC-PP | nsamples=4000, nwalkers=50, burn-in=50, thin=100 | 1.629 (0.01) | 1.943 (0.011) | 2.266 (0.009) | 2.467 (0.009) | 26.672 (7.3E-01) |
| MCMC-PP | nsamples=4000, nwalkers=50, burn-in=100, thin=100 | 1.631 (0.01) | 1.943 (0.011) | 2.267 (0.009) | 2.469 (0.009) | 26.997 (7.4E-01) |
| MCMC-PP | nsamples=4000, nwalkers=50, burn-in=50, thin=10 | 1.631 (0.01) | 1.944 (0.011) | 2.267 (0.009) | 2.468 (0.009) | 29.109 (6.3E-01) |
| MCMC-PP | nsamples=4000, nwalkers=50, burn-in=50, thin=1 | 1.631 (0.01) | 1.944 (0.011) | 2.267 (0.009) | 2.469 (0.009) | 53.601 (4.2E-01) |
| MCMC-PP | nsamples=4000, nwalkers=50, burn-in=100, thin=10 | 1.632 (0.01) | 1.943 (0.011) | 2.268 (0.01) | 2.47 (0.009) | 29.433 (6.4E-01) |
| MCMC-PP | nsamples=4000, nwalkers=50, burn-in=100, thin=1 | 1.632 (0.01) | 1.943 (0.011) | 2.268 (0.01) | 2.47 (0.009) | 53.941 (4.2E-01) |
| MCMC-PP | nsamples=4000, nwalkers=50, burn-in=500, thin=100 | 1.632 (0.01) | 1.94 (0.012) | 2.274 (0.01) | 2.477 (0.009) | 29.601 (8.2E-01) |
| MCMC-PP | nsamples=4000, nwalkers=50, burn-in=500, thin=10 | 1.632 (0.01) | 1.942 (0.011) | 2.275 (0.01) | 2.477 (0.009) | 32.036 (7.1E-01) |
| MCMC-PP | nsamples=4000, nwalkers=50, burn-in=500, thin=1 | 1.632 (0.01) | 1.942 (0.011) | 2.275 (0.01) | 2.477 (0.009) | 56.526 (3.4E-01) |
| MCMC-PP | nsamples=4000, nwalkers=100, burn-in=0, thin=100 | 1.637 (0.01) | 1.954 (0.011) | 2.274 (0.01) | 2.474 (0.009) | 44.076 (1.4E+00) |
| MCMC-PP | nsamples=4000, nwalkers=100, burn-in=0, thin=10 | 1.639 (0.01) | 1.956 (0.011) | 2.276 (0.01) | 2.476 (0.009) | 48.947 (1.2E+00) |
| MCMC-PP | nsamples=4000, nwalkers=100, burn-in=0, thin=1 | 1.639 (0.01) | 1.956 (0.011) | 2.276 (0.01) | 2.476 (0.009) | 98.033 (9.9E-01) |
| MCMC-PP | nsamples=4000, nwalkers=100, burn-in=50, thin=100 | 1.639 (0.01) | 1.956 (0.011) | 2.276 (0.01) | 2.476 (0.009) | 44.618 (1.4E+00) |
| MCMC-PP | nsamples=4000, nwalkers=100, burn-in=50, thin=10 | 1.64 (0.01) | 1.958 (0.011) | 2.277 (0.01) | 2.477 (0.009) | 49.475 (1.2E+00) |
| MCMC-PP | nsamples=4000, nwalkers=100, burn-in=50, thin=1 | 1.64 (0.01) | 1.958 (0.011) | 2.277 (0.01) | 2.477 (0.009) | 98.548 (1.0E+00) |
| MCMC-PP | nsamples=4000, nwalkers=100, burn-in=100, thin=100 | 1.641 (0.01) | 1.958 (0.011) | 2.277 (0.01) | 2.477 (0.009) | 45.16 (1.4E+00) |
| MCMC-PP | nsamples=4000, nwalkers=100, burn-in=100, thin=10 | 1.641 (0.01) | 1.957 (0.011) | 2.278 (0.01) | 2.478 (0.009) | 50.011 (1.2E+00) |
| MCMC-PP | nsamples=4000, nwalkers=100, burn-in=100, thin=1 | 1.641 (0.01) | 1.957 (0.011) | 2.278 (0.01) | 2.478 (0.009) | 98.98 (1.1E+00) |
| MCMC-PP | nsamples=4000, nwalkers=100, burn-in=500, thin=100 | 1.642 (0.01) | 1.955 (0.011) | 2.284 (0.01) | 2.485 (0.009) | 49.5 (1.6E+00) |
| MCMC-PP | nsamples=4000, nwalkers=100, burn-in=500, thin=10 | 1.642 (0.01) | 1.956 (0.011) | 2.285 (0.01) | 2.486 (0.009) | 54.328 (1.3E+00) |
| MCMC-PP | nsamples=4000, nwalkers=100, burn-in=500, thin=1 | 1.642 (0.01) | 1.956 (0.011) | 2.285 (0.01) | 2.486 (0.009) | 103.151 (1.1E+00) |
| LC-PFN | Nb data=100k, nlayers=6, emsize=128 | 1.242 (0.001) | 1.508 (0.001) | 1.652 (0.001) | 1.709 (0.002) | 0.007 (4.6E-04) |
| LC-PFN | Nb data=100k, nlayers=12, emsize=128 | 1.254 (0.001) | 1.522 (0.001) | 1.669 (0.001) | 1.73 (0.002) | 0.012 (8.3E-04) |
| LC-PFN | Nb data=100k, nlayers=6, emsize=256 | 1.267 (0.002) | 1.597 (0.001) | 1.764 (0.001) | 1.843 (0.002) | 0.011 (8.2E-04) |
| LC-PFN | Nb data=100k, nlayers=12, emsize=256 | 1.384 (0.001) | 1.664 (0.001) | 1.833 (0.001) | 1.9 (0.002) | 0.021 (1.6E-03) |
| LC-PFN | Nb data=100k, nlayers=3, emsize=512 | 1.392 (0.001) | 1.679 (0.001) | 1.871 (0.001) | 1.959 (0.002) | 0.016 (1.4E-03) |
| LC-PFN | Nb data=100k, nlayers=6, emsize=512 | 1.439 (0.001) | 1.745 (0.001) | 1.947 (0.001) | 2.034 (0.002) | 0.028 (2.5E-03) |
| LC-PFN | Nb data=100k, nlayers=12, emsize=512 | 1.504 (0.001) | 1.817 (0.001) | 2.013 (0.001) | 2.091 (0.002) | 0.048 (4.2E-03) |
| LC-PFN | Nb data=1M, nlayers=3, emsize=128 | 1.528 (0.001) | 1.894 (0.001) | 2.159 (0.001) | 2.279 (0.002) | 0.004 (2.8E-04) |
| LC-PFN | Nb data=1M, nlayers=6, emsize=128 | 1.581 (0.001) | 1.949 (0.001) | 2.202 (0.001) | 2.308 (0.002) | 0.007 (4.6E-04) |
| LC-PFN | Nb data=1M, nlayers=3, emsize=256 | 1.583 (0.001) | 1.967 (0.001) | 2.239 (0.001) | 2.357 (0.002) | 0.007 (5.0E-04) |
| LC-PFN | Nb data=10M, nlayers=3, emsize=128 | 1.585 (0.001) | 1.989 (0.001) | 2.282 (0.002) | 2.431 (0.003) | 0.004 (2.8E-04) |
| LC-PFN | Nb data=1M, nlayers=12, emsize=128 | 1.621 (0.001) | 1.982 (0.001) | 2.223 (0.001) | 2.326 (0.002) | 0.012 (8.2E-04) |
| LC-PFN | Nb data=1M, nlayers=3, emsize=512 | 1.624 (0.001) | 2.011 (0.001) | 2.291 (0.001) | 2.417 (0.002) | 0.016 (1.4E-03) |
| LC-PFN | Nb data=1M, nlayers=6, emsize=256 | 1.627 (0.001) | 2.007 (0.001) | 2.269 (0.001) | 2.381 (0.002) | 0.012 (8.7E-04) |
| LC-PFN | Nb data=10M, nlayers=3, emsize=256 | 1.654 (0.001) | 2.044 (0.001) | 2.347 (0.002) | 2.493 (0.003) | 0.006 (4.7E-04) |
| LC-PFN | Nb data=1M, nlayers=12, emsize=256 | 1.667 (0.001) | 2.035 (0.001) | 2.294 (0.001) | 2.406 (0.002) | 0.02 (1.5E-03) |
| LC-PFN | Nb data=1M, nlayers=6, emsize=512 | 1.672 (0.001) | 2.051 (0.001) | 2.323 (0.001) | 2.441 (0.002) | 0.028 (2.4E-03) |
| LC-PFN | Nb data=10M, nlayers=6, emsize=128 | 1.673 (0.001) | 2.058 (0.001) | 2.352 (0.001) | 2.478 (0.003) | 0.007 (4.6E-04) |
| LC-PFN | Nb data=1M, nlayers=12, emsize=512 | 1.699 (0.001) | 2.072 (0.001) | 2.344 (0.001) | 2.46 (0.002) | 0.051 (4.6E-03) |
| LC-PFN | Nb data=10M, nlayers=3, emsize=512 | 1.707 (0.001) | 2.083 (0.001) | 2.378 (0.002) | 2.517 (0.003) | 0.017 (1.4E-03) |
| LC-PFN | Nb data=10M, nlayers=12, emsize=128 | 1.717 (0.001) | 2.094 (0.001) | 2.377 (0.001) | 2.498 (0.003) | 0.012 (8.1E-04) |
| LC-PFN | Nb data=10M, nlayers=6, emsize=256 | 1.721 (0.001) | 2.098 (0.001) | 2.387 (0.002) | 2.514 (0.003) | 0.011 (8.2E-04) |
| LC-PFN | Nb data=10M, nlayers=6, emsize=512 | 1.744 (0.001) | 2.125 (0.001) | 2.404 (0.002) | 2.527 (0.003) | 0.026 (2.3E-03) |
| LC-PFN | Nb data=10M, nlayers=12, emsize=256 | 1.746 (0.001) | 2.124 (0.001) | 2.403 (0.001) | 2.522 (0.003) | 0.02 (1.5E-03) |
| LC-PFN | Nb data=10M, nlayers=12, emsize=512 | 1.758 (0.001) | 2.133 (0.001) | 2.403 (0.002) | 2.519 (0.003) | 0.05 (4.5E-03) |

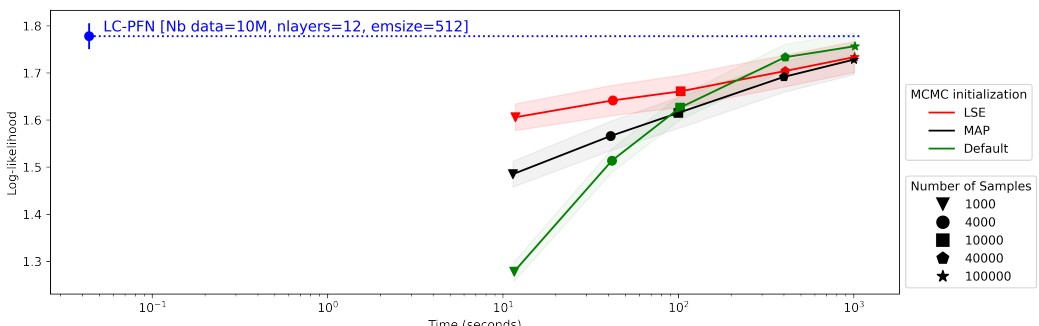

Figure 10: runtime *vs* log-likelihood values of `MCMC-PP`, using thinning 100, with different initialization strategies and sample sizes (per walker), averaged across 1 000 curves generated i.i.d. from the prior. The blue point represents `LC-PFN`, with the dotted line as a reference for its LL value. The shaded areas correspond to ± 1 standard error.

### C.1.2  Effect of MCMC chain length and initialization

In our experiments in Section 4.1 we observed that the quality of MCMC inference increases with chain length. In particular, our best `MCMC-PP` variant (`M3` in Figure 3), uses the same hyperparameter settings as Domhan et al. [2015] (`M1`), but with a double as long chain (discounting the burn-in). In this section, we investigate whether the chains considered are too short and whether increasing chain lengths further will eventually cause `MCMC-PP` to outperform `LC-PFN`. To study this, we will run `MCMC-PP` to collect up to 100 000 samples per walker (50× more than Domhan et al. [2015]). To reduce the computational cost of this experiment, we only extrapolate 1 000 (instead of 10 000) prior curves, on cutoff 10%, and use `MCMC-PP` with a 100-thinning strategy (`M2`). This variant is more compute-efficient and the minor negative effect of thinning is expected to further decrease with increasing chain length. Concurrently, we study the effect of initialization. There are different ways of initializing MCMC chains and the chosen strategy can strongly affect performance if chains are too short. Domhan et al. [2015] initializes the chain by setting the parameters for each of the $K$ basis curves to their Least-Squared Estimates (`LSE`). Weights are initialized to be $\frac{1}{K}$. If this initial point violates the constraints imposed by the prior, a default starting point is used instead. As an ablation, we compare it to an initialization always starting at the default (`default`). As another ablation, we compare to a strategy using the maximum a posteriori estimate (`MAP`) as a starting point instead, which unlike LSE takes the likelihood under the prior into account.

Figure 10 shows the quality and cost of inferences for each of the initialization strategies when collecting up to 100 000 samples (per walker). While these results clearly show that increasing chain length continues to improve performance. The trends we observe, suggest that `MCMC-PP` could eventually attain or even overtake the best `LC-PFN`. That being said, the best MCMC we considered has 20 000× longer runtimes than that `LC-PFN` requires, yet it does not quite reach the same performance, so outperforming it would require impractically long chains. When optimistically extending the trends, we estimate to need at least 10× longer chains, and inference times of multiple hours. In terms of initialization, we observe that MCMC performance for shorter chains indeed more strongly depends on the initialization strategy, where the `LSE` strategy by Domhan et al. [2015] does best for short chains, followed by `MAP` and `default`. When increasing chain length, we find that differences get smaller and order reverses such that the fixed `default` initialization is best, suggesting that greedy initialization (`LSE`, `MAP`) may hurt performance on some curves.

### C.2  Extrapolating real-world learning curves

In addition to the average rank plots presented in Section 4.2, we also perform pairwise comparisons of the log-likelihood and mean squared error values among the different methods (`LC-PFN` and `MCMC` variants). This approach allows for a more detailed analysis, as it compares the absolute values of the metrics. Unlike the rank plots, these pair and curvewise plots provide insights into outliers which thus further enhance our understanding of the results.

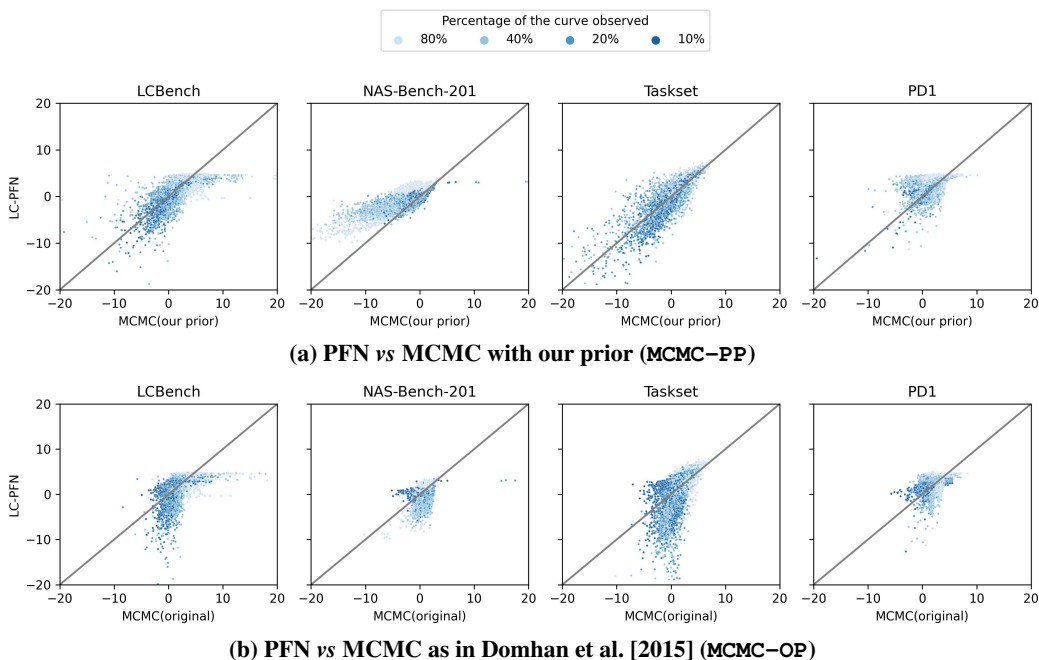

**(a) PFN *vs* MCMC with our prior (`MCMC-PP`)**

**(b) PFN *vs* MCMC as in Domhan et al. [2015] (`MCMC-OP`)**

Figure 11: Pairwise comparison of log-likelihood (higher is better) between PFN and MCMC.

Figure 11 presents a pairwise comparison of the log-likelihood values between our PFN method and the two variants of MCMC, per curve, considering different cutoff values on the four benchmark datasets. Figure 12 presents the same comparison based on mean squared error (MSE).

We observe that `LC-PFN` compares favorably to `MCMC-PP` on `PD1` and `NAS-Bench-201` tasks. Both methods demonstrate comparable performance on the `Taskset` dataset. However, it is worth noting that for some curves of the `LCBench` dataset, `MCMC-PP` obtains very high log-likelihood and low MSE scores, while `LC-PFN` falls short in this aspect. We believe this discrepancy can be attributed to the fact that the `LCBench` dataset contains a significant number of constant curves (see Figure 8a) that fall within the same bin of the discretized PPD, effectively limiting the the maximal confidence / accuracy of `LC-PFN`. Looking at MSE specifically, we find that for some `LCBench` curves, at high cutoffs, `LC-PFN` obtains very high errors, while `MCMC-PP` does not. Upon closer inspection, we found that these curves quickly converge to a value close to the optimal accuracy, and while `LC-PFN` captures this trend for lower cutoffs, it suddenly fails for larger cutoffs (see 5th example in Figure 18). This can likely be explained by the fact that these curves are not adequately captured by the prior proposed in Section 3.2 and this seems to be one of the few cases where `LC-PFN` generalizes poorly out of distribution.

When comparing with `MCMC-OP`, we observe that `MCMC-OP` has relatively few outliers in terms of log-likelihood, both in the positive and negative sense. This can likely be explained by the prior of `MCMC-OP`, which is more flexible (i.e. has higher entropy) than ours and making `MCMC-OP` thus less confident about its predictions. However, `MCMC-OP` performs clearly worse in terms of MSE, suggesting that the median of the high entropy PPDs produced by this method do not provide a good point estimate.

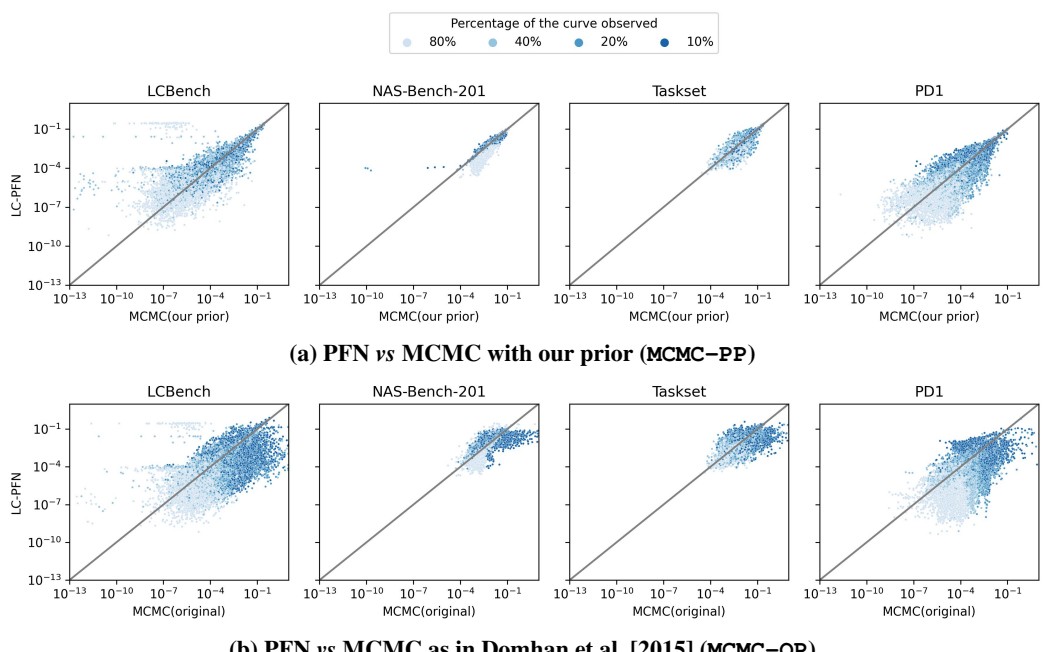

**(a) PFN *vs* MCMC with our prior (`MCMC-PP`)**

**(b) PFN *vs* MCMC as in Domhan et al. [2015] (`MCMC-OP`)**

Figure 12: Pairwise comparison of MSE values (lower is better) between PFN and MCMC.

## C.3 Application: Extrapolation-based early stopping in model selection

In what follows, we further analyze and discuss our experiments described in Section 4.3.

### C.3.1 Results on individual tasks

While Figure 5 shows the results of our early stopping experiments for each `PD1` task, results for the three remaining benchmarks are summarized by averaging them across all tasks. Since this may hide task-dependent variation and outliers, we have a closer look at the performance on individual tasks. Figure 13 shows the results for each of 35 `LCBench` tasks. Here, we observe that relative performances vary per task. The aggressive `Patience(k)` (i.e., low $k$) baselines perform best on some tasks, but fail on others. The `LC-PFN` based termination criteria consistently perform well on each task, obtaining a 2-6× speed-up (w.r.t. the final performance of `no-stop`) on 30 of the 35 tasks, and no significant slow-down on the remaining five. Figure 14 shows the results for each of the 3 `NAS-Bench-201` tasks. Here, we observe failure on all 3 tasks, where the degree of failure seems correlated with the complexity of the image classification task. Figure 15 shows the results for each of the 12 `Taskset` tasks. Here, the `LC-PFN` based termination criteria consistently perform well, obtaining 2-6× speed-ups on all 12 tasks. We conclude that Figure 5 accurately reflects relative performances on all four benchmarks.

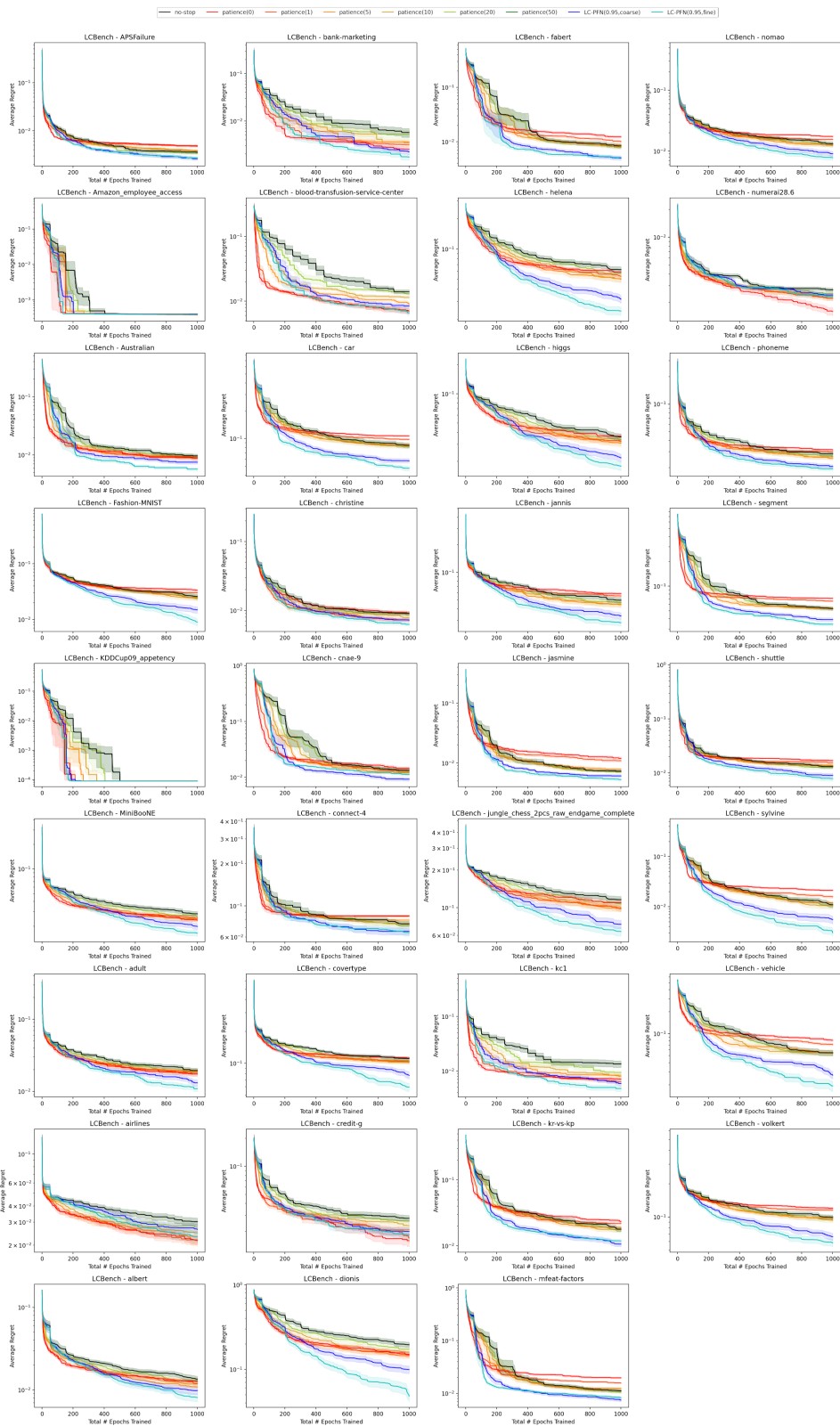

Figure 13: Average anytime regret (lower is better) obtained by all early stopping criteria for 40 run orderings for each of the 35 `LCBench` tasks.

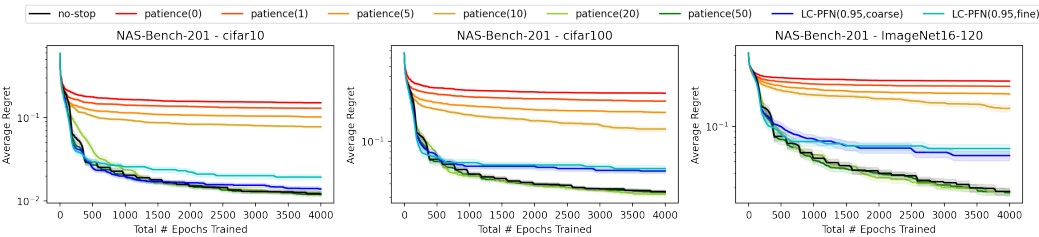

Figure 14: Average anytime regret (lower is better) obtained by all early stopping criteria for 40 run orderings for each of the 3 `NAS-Bench-201` tasks.

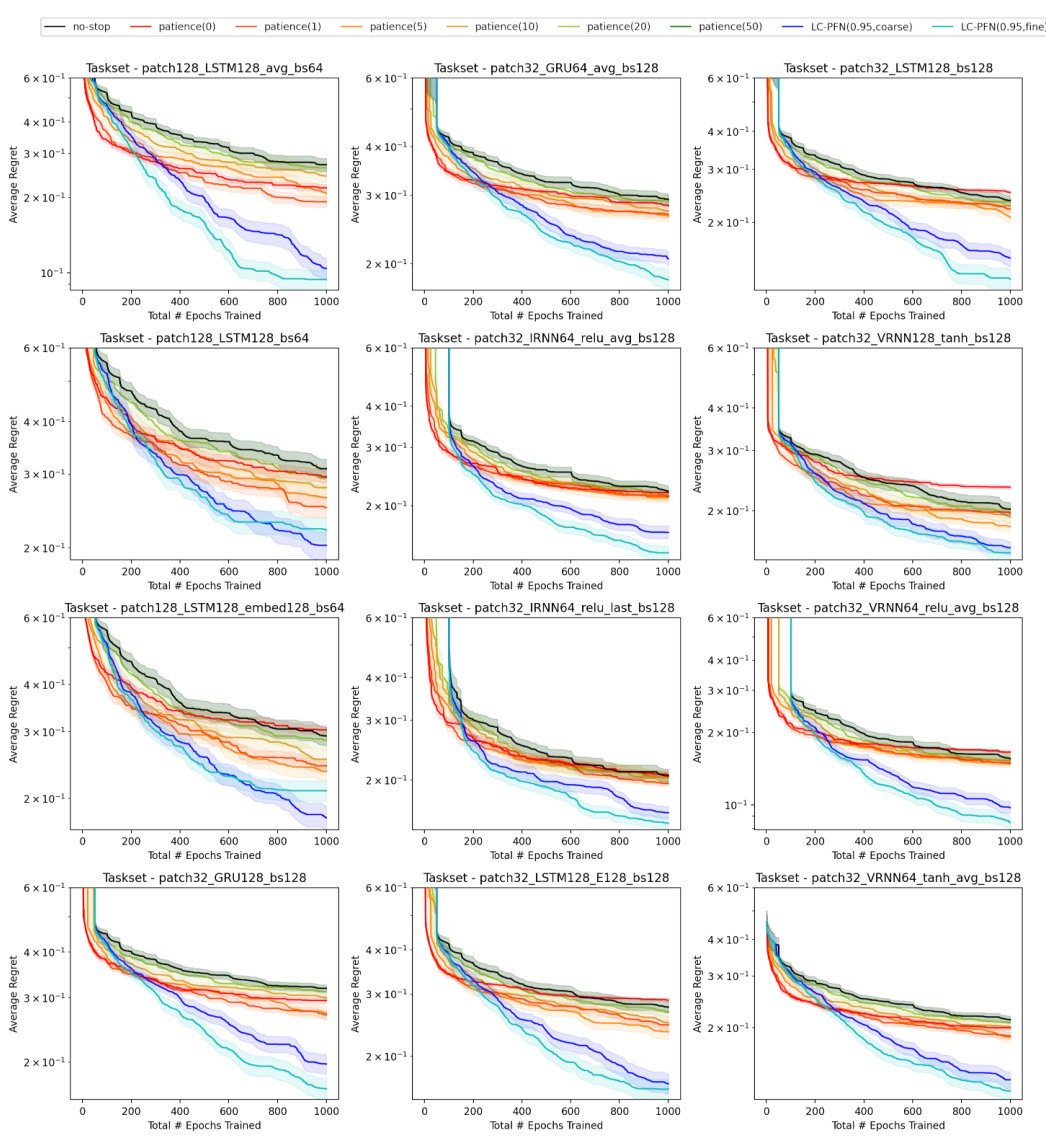

Figure 15: Average anytime regret (lower is better) obtained by all early stopping criteria for 40 run orderings for each of the 12 `Taskset` tasks.

### C.3.2 Parameter sensitivity analysis

In our comparison in Figure 5, we considered two variants of the `LC-PFN` based termination criterion, varying the frequency at which it is applied, and found the fine-grained strategy, applying it every epoch, to perform best. However, many variants of this scheme exist, and in what follows we investigate the impact of some of our other choices on performance.

**Confidence level:** In our experiments, we terminate a training run if we are confident it will not improve upon the best model seen so far. Following prior art [Domhan et al., 2015], we adopted a confidence threshold of 0.95. Figure 16 (a) shows the anytime performance of variants using lower (0.90) and higher (0.99, 0.999) confidence levels. We observe that all of these perform relatively well, suggesting some degree of robustness. The speed-ups obtained using higher confidence levels are mostly lower (up to $2\times$), but more consistent. In particular, we only observe minimal slow-down on `NAS-Bench-201` at confidence level 0.999, and higher confidences are also beneficial for the `PD1` protein sequence (UniRef50) task.

**Minimal cutoff:** While we apply our predictive termination criterion every epoch, we do not apply it "as of the first epoch", but only after two observations are available ($T \geq 2$). Since `LC-PFN`'s predictions are conditioned on the partial curve only, it will predict the prior for $T = 0$. Beyond not allowing us to discriminate between runs, it introduces a potential failure mode for easy tasks (many training runs obtain models close to the theoretical optimum) and `LC-PFN` (unaware of task hardness) will terminate runs before they even began because it is very unlikely to obtain a better model *under the prior*. Figure 16 (b) shows the anytime performance of variants that apply the criterion as soon as one, three, four, or five observations are made. We find that a choice of one (instead of two) only negatively impacts performance on `LCBench` (containing some very easy tasks). A minimal cutoff of two works well on all benchmarks, with minor slow-downs for the higher minimal cutoffs.

### C.4 Qualitative plots of learning curve extrapolation

To complement the quantitative evaluation in the main paper, we provide examples of extrapolations, at the four different cutoffs, for seven curves from the prior (Figure 17), `LCBench` (Figure 18), `NAS-Bench-201` (Figure 19), `Taskset` (Figure 20), and `PD1` (Figure 21). The plots show the median and two-sided 90% confidence interval of the PPDs inferred using the `LC-PFN` and `MCMC-PP` variant considered in Section 4.2. We observe that while the quality of the extrapolations produced by `LC-PFN` varies strongly, they are mostly logical, given the data observed and the prior used, and we find that the inferences of `MCMC-PP` are rarely preferable.

## D Breakdown computational cost

Overall, reproducing all our experiments in the main paper requires approximately 163 CPU days and 60 GPU hours on our systems (GPU: NVIDIA (R) GeForce (R) RTX 2080, CPU: Intel(R) Xeon(R) CPU E5-2630 v4 @ 2.20GHz). These costs break down as follows:

- **Section 4.1** entails 80 CPU days to run all `MCMC-PP` variants on the 10 000 sampled curves. The overall prediction time of `LC-PFN` is negligible (<1 hr for each variant). However, for `LC-PFN`, there is an initial training cost, which amounts to approximately 60 GPU hours to train all 27 `LC-PFN` variants.

- **Section 4.2** requires 80 CPU days to run the MCMC algorithm on the considered real curve benchmarks.

- **Section 4.3** involves a maximum of 3 CPU days to replicate the early stopping results on all the benchmarks for both `LC-PFN` and the baselines.

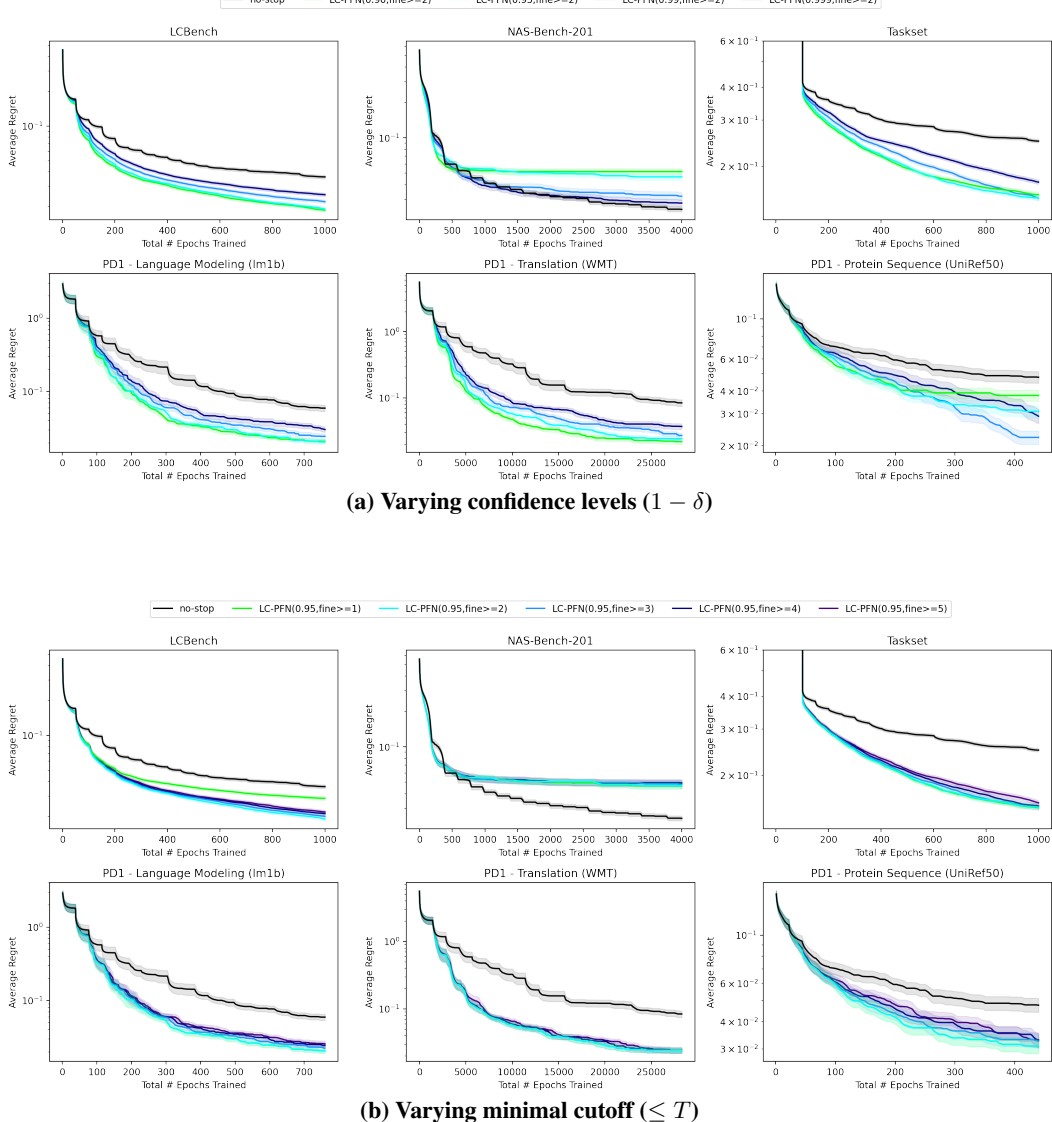

Figure 16: Parameter sensitivity analysis of our fine-grained `LC-PFN` based termination criterion in terms of the average regret (lower is better).

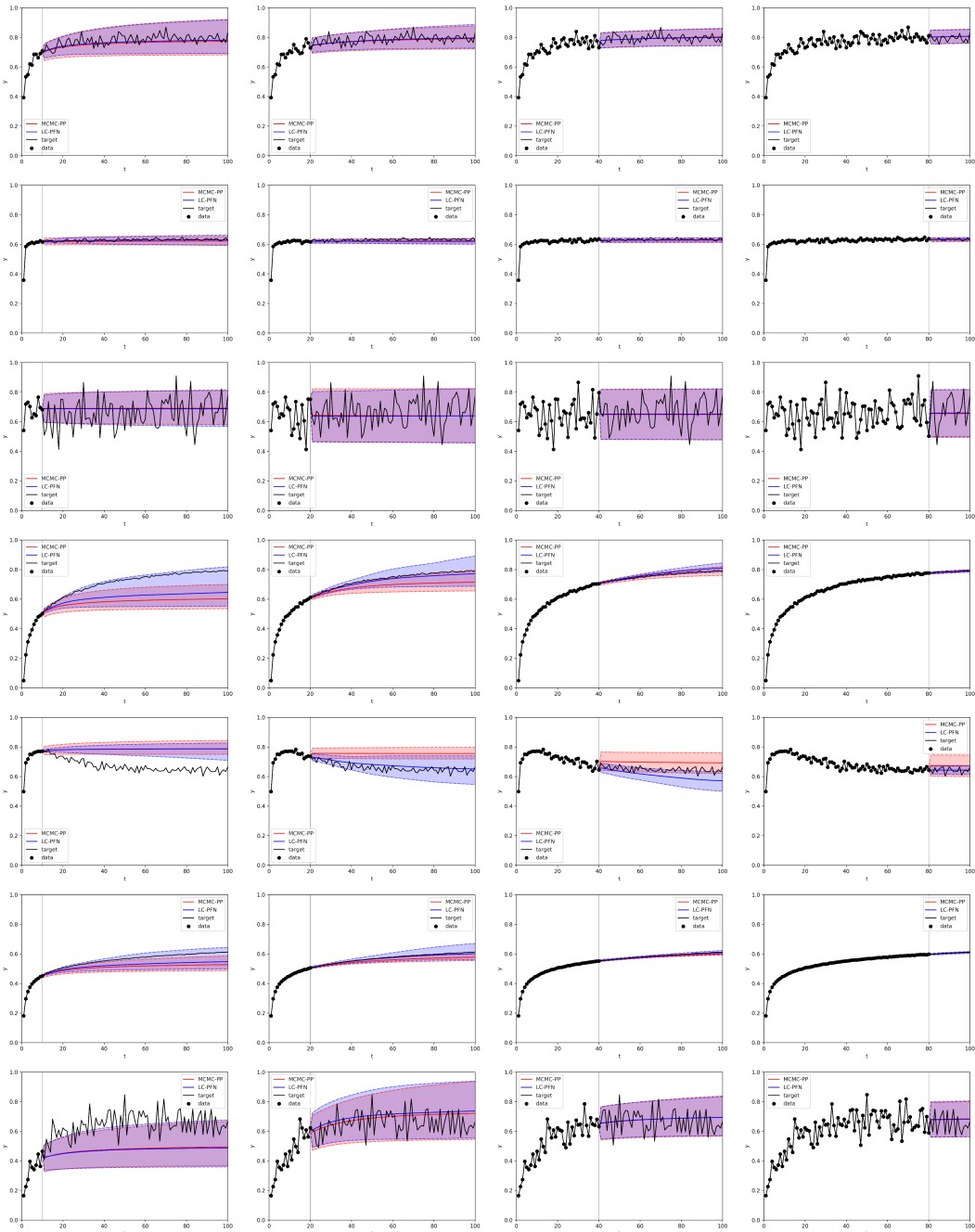

Figure 17: Extrapolations of 7 different curves from the prior at 10, 20, 40, and 80 cutoff

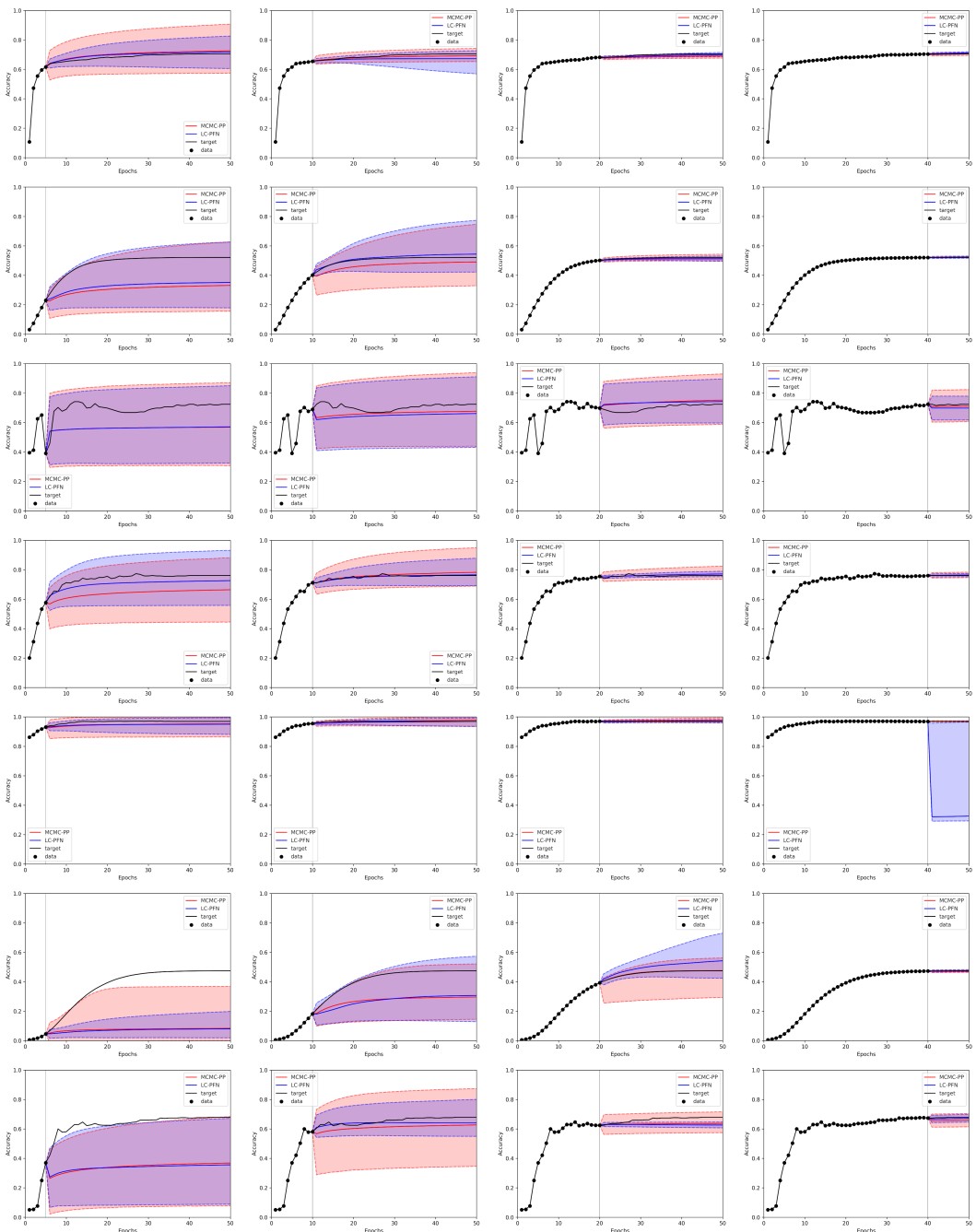

Figure 18: Extrapolations of 7 different curves from `LCBench` at 10%, 20%, 40%, and 80% cutoff

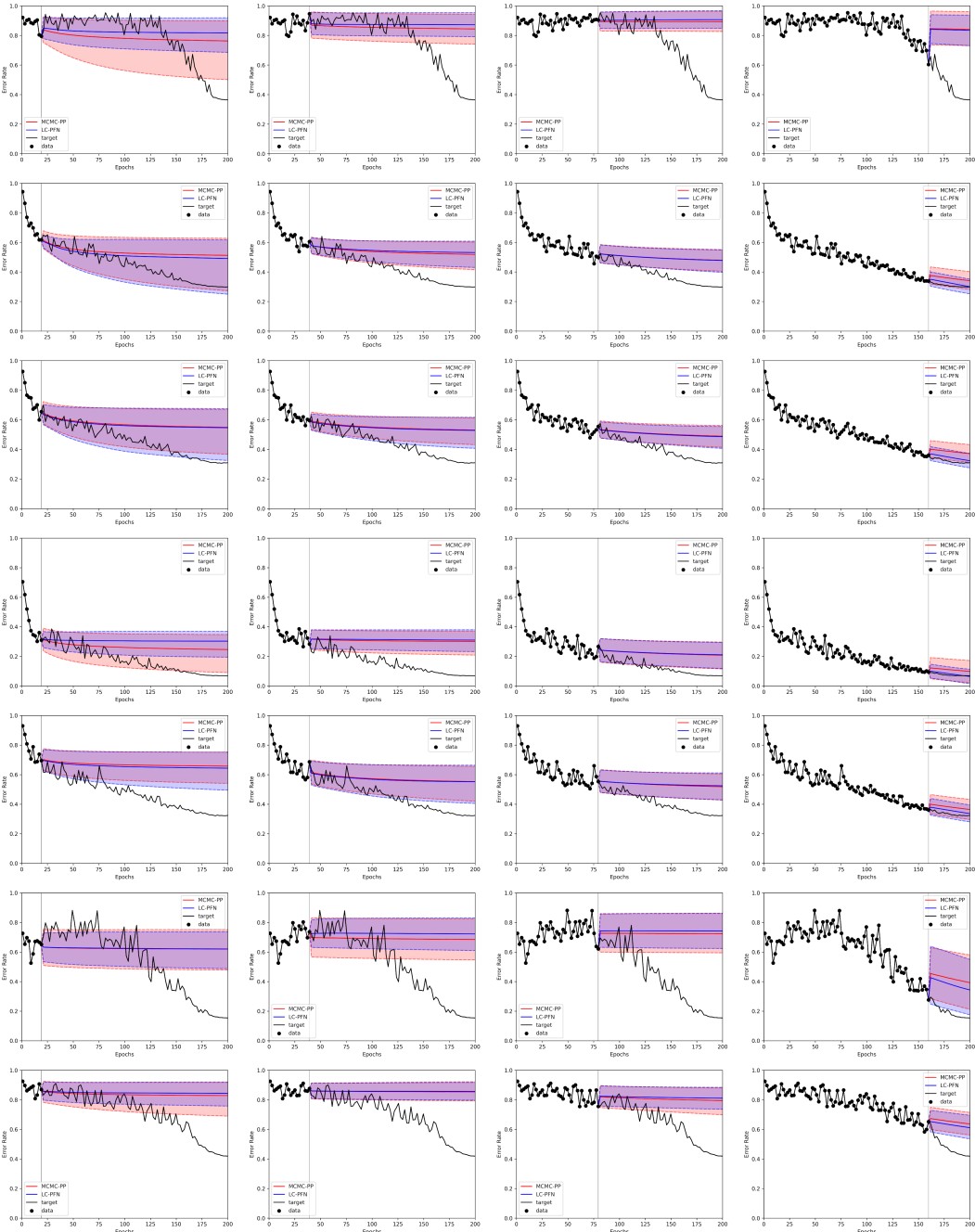

Figure 19: Extrapolations of 7 different curves from `NAS-Bench-201` at 10%, 20%, 40%, and 80% cutoff

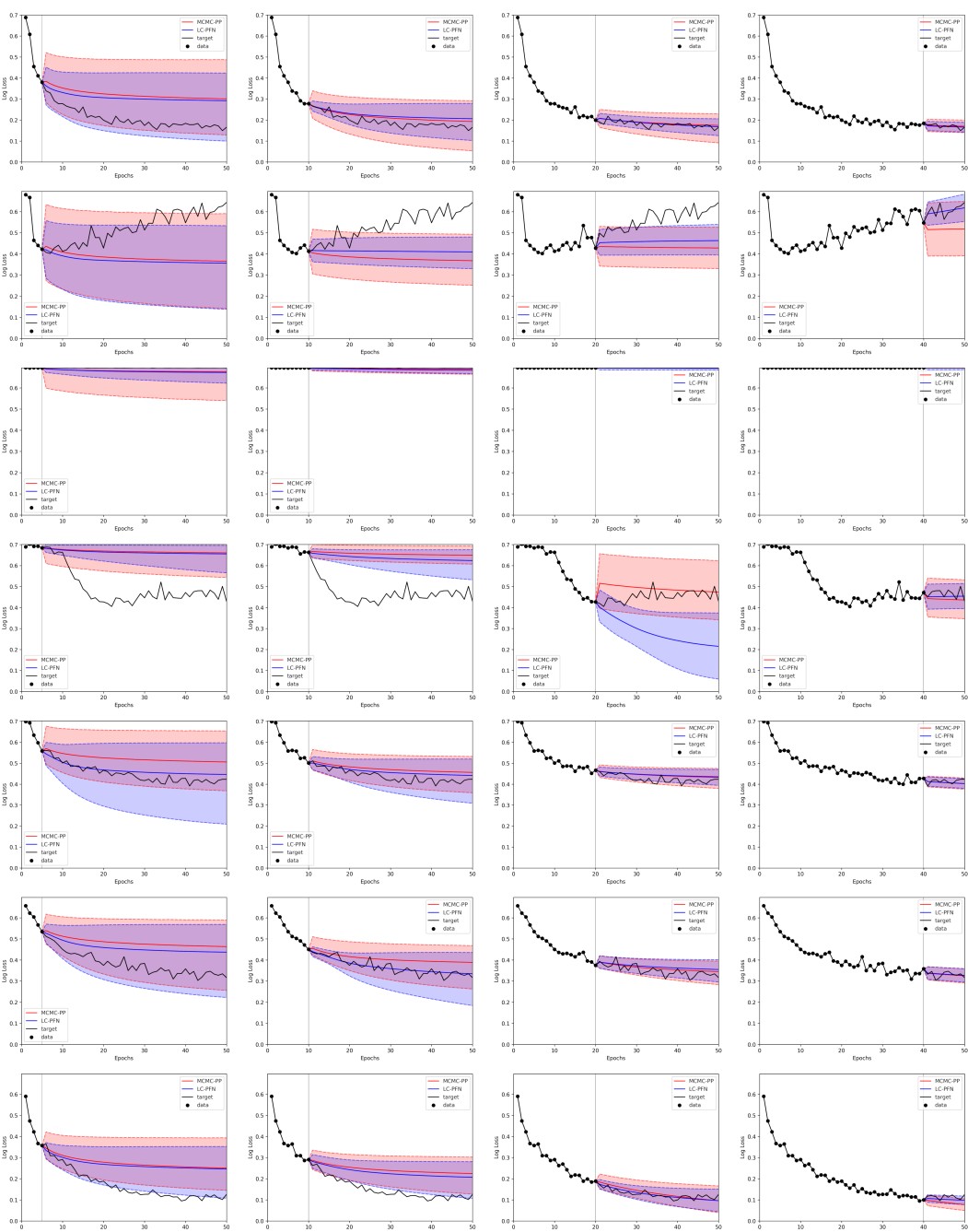

Figure 20: Extrapolations of 7 different curves from `Taskset` at 10%, 20%, 40%, and 80% cutoff

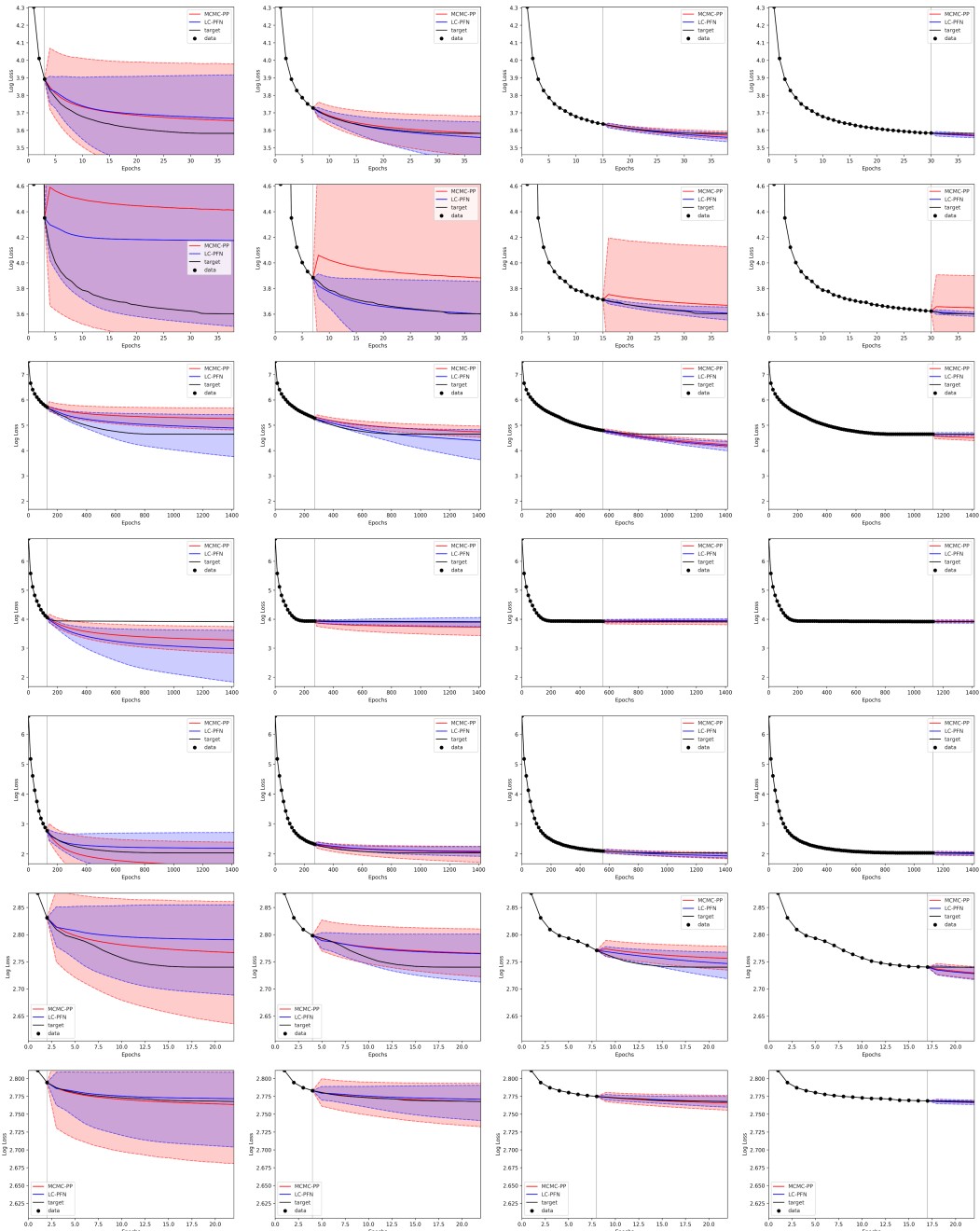

Figure 21: Extrapolations of 7 different curves from `PD1` at 10%, 20%, 40%, and 80% cutoff

