# OpenReview forum: "Efficient Bayesian Learning Curve Extrapolation using Prior-Data Fitted Networks"
_NeurIPS.cc/2023/Conference — NeurIPS 2023 poster_

### Official Review · Reviewer_3hos · 2023-06-16

**Soundness:** 3 good
**Presentation:** 3 good
**Contribution:** 3 good
**Rating:** 7
**Confidence:** 3

**Summary:**

The authors consider the task of learning curve extrapolation, i.e., the aim is to predict the performance of a given model wrt. e.g., accuracy/log-likelihood over time, given current observations.
Their proposal primarily relies on Prior-data fitted networks (Müller et al., 2022) trained on samples from a prior set of curves adapted from their sole comparison method by Dunham et al. (2015).
Compared to this prior MCMC-based method (Dunham et al., 2015), the authors can show greatly improved performance in most setups as well as very strong improvements in runtime.


**Strengths:**

- The paper focuses on an important task in the AutoML literature that has so far not received a lot of attention.
- The authors evaluate their approach extensively on a varied set of experiments properly evaluating all their claims.
- The paper is overall well written with two minor deficiencies (see next section).
- The authors provide an extensive implementation of the model. However, it is also necessary as replicability would be difficult given the coarse level of detail in the written part.
- With respect to originality. The contribution is primarily in the application. The model itself relies on the PFN model by Mueller and the prior is an adaptation of Dunham et al.
This can be seen as a weakness, yet given that the application of PFNs to this field is novel and the application is an important one with clear results, I see this only as a very minor weakness if at all.
- Concerning significance, I lack a deep understanding of the AutoML literature to properly judge the significance. But from what I know of the field, the results look very promising and should be of great interest to many readers.

**Weaknesses:**

- It has two deficiencies in the writing/structure of the paper.
   1.  Its backbone is a prior-data fitted network. However, that model itself is only briefly introduced in a single paragraph with a figure that is barely understandable without reading the original paper.
The paper requires a proper discussion of this approach in either the main text or the appendix.
   2. Training details and hyperparameters are barely discussed with many essentials missing, e.g., the reader has to guess the meaning of `nb_data`, `emsize`, `nlayers`. It is an easy guess, but a guess nevertheless. Adding the abbreviations to the paragraph in l157 would quickly fix this.
- As stated above, the novelty is almost solely in the application which could be considered a weakness. But in my opinion this is only a very minor weakness if at all.
- Table 3 lacks error bars.

## Minor
- Figure 2 is missing details on the colors of the arrows
- The caption of Table two should be above
- Figure 3 contains an unexplained horizontal line
- l319-320 the sentence is broken (interesting appears twice)

**Questions:**

- In l216 the authors mention that the best MCMC-PP method took over 80 seconds. In Table 3 the 80s method does, however, not not improve upon the 30-second version. Can the authors clarify this statement?


**Limitations:**

The authors discuss limitations of their method, but no potential negative societal impact of their work. The second is of lesser importance in this work anyway.

---

> ### Author Rebuttal · Authors · 2023-08-10
>
> ### Regarding the weaknesses you raised:
>
> **Its backbone is a prior-data fitted network. However, that model itself is only briefly introduced in a single paragraph with a figure that is barely understandable without reading the original paper. The paper requires a proper discussion of this approach in either the main text or the appendix.**
> We agree that our description thereof in the current manuscript is rather high-level. We will add a section elaborating on some PFN-specific implementation details and design choices in the appendix. (We also would like to mention that the code for both PFN and LC-PFN is open source, at least specifying all details in the code. But that’s of course not an excuse for the method to be only explained very briefly in the paper, and we’ll fix that.)
>
> **Training details and hyperparameters are barely discussed with many essentials missing, e.g., the reader has to guess the meaning of ``nb_data``, ``emsize``, ``nlayers``. It is an easy guess, but a guess nevertheless. Adding the abbreviations to the paragraph in l157 would quickly fix this.**
>
> We will modify this paragraph accordingly, thank you for pointing out this issue and your suggestion.
>
> **As stated above, the novelty is almost solely in the application which could be considered a weakness. But in my opinion this is only a very minor weakness if at all.**
>
> Agreed, thank you for this perspective.
>
> **Table 3 lacks error bars.**
> Our apologies, please find the updated table including the standard error in the PDF attached to the global rebuttal (as Table 1).
>
> **Minor comments**
> We will make the necessary changes, thank you for pointing out these minor issues.
>
>
> ### Regarding your questions / comments:
>
> **In l216 the authors mention that the best MCMC-PP method took over 80 seconds. In Table 3 the 80s method does, however, not not improve upon the 30-second version. Can the authors clarify this statement?**
>
> This is an unfortunate error in the discussion. Our apologies and thank you for spotting it.
> It should say  “the best MCMC-PP method took over 30 seconds.”
>
> Context: In our preliminary experiment, conducted with a smaller sample size, the mentioned MCMC configuration appeared to show a slight advantage. However, this difference did not hold up in the final, large-scale experiment. Unfortunately, we overlooked updating this information in our discussion, and we apologize for any confusion this may have caused. In the final experiment, in Section 4.1, we found that no single MCMC variant consistently outperformed others across different cutoffs. Given the subtle differences among top-performing configurations and the varying computational efficiency, we chose the more efficient variant for future comparisons. Note that this ‘cheaper’ MCMC variant still requires multiple orders of magnitude time than the cheapest LC-PFN variant (to obtain a worse average LL).

---

> > ### Comment · Reviewer_3hos · 2023-08-16
> >
> > Thank you for your further clarifications. I keep my score of recommending acceptance.

---

### Official Review · Reviewer_7zKX · 2023-07-06

**Soundness:** 2 fair
**Presentation:** 3 good
**Contribution:** 2 fair
**Rating:** 5
**Confidence:** 3

**Summary:**

The authors in this submission applied the prior-data fitted neural networks (PFNs) in the learning curve extrapolation task, for which the main goal is to predicit the performance of a machine learning model in later epochs, based on the information from earlier epochs. The target is modeled as a linear combination of basis growth curves, and the proposed LC-PFN algorithm aims to minimize the cross entropy loss as in PFN. The authors tested the proposed method across a few datasets and showed that PFN (i) achieves better prediction and (ii) is computationally efficient, when compared with its competitors.

**Strengths:**

1. The idea of applying PFNs in the learning curve extrapolation task sounds reasonable, and is intuitively more efficient when compared with MCMC.

2. The application of LC-PFN in early stopping could be valuable, as it can be helpful for the model selection step. The experiments in Section 4.3 also provide some promising results.

**Weaknesses:**

1. In my opinion the selection for the 3 parametric basis curves seems a bit ad-hoc. It's not convinving whether they are sufficient to fit different learning curves. Furthermore, selection of hyperparemters in priors there also lacks details.

2. The results could be more convincing if the authors can compare aganist more previous methods, instead of MCMC only. For instance, how about the one by Klein et al., 2017, which used Bayesian neural networks for learning curve prediction?

**Questions:**

1. In Table 1, is the support for x integers, as it corresponds to $f_k (t | \theta_k)$ in the model?

2. For the equation just below L132, what does $<$ mean?

3. Do you need to assume $t > T^{'}$ in your LC-PFN model during training?

**Limitations:**

Yes

---

> ### Author Rebuttal · Authors · 2023-08-10
>
> ### Regarding the weaknesses you raised:
>
> **In my opinion the selection for the 3 parametric basis curves seems a bit ad-hoc. It's not convincing whether they are sufficient to fit different learning curves. Furthermore, selection of hyperparameters in priors there also lacks details.**
>
> Thanks for the question. We would like to add some further context on how we got to this prior: For a fair comparison, we chose this prior to mimic the setting in Domhan et al. (2015), which is the most closely related to the prior art. Sadly, using the exact same prior was not possible (we cannot generate samples from a uniform distribution with unbounded support, see Section 3.2). Hence, we decided to ‘peak’ the prior / limit parameter ranges to cover a wide spectrum of possible curves, while still guaranteeing the competitiveness of our baseline. Despite these efforts, we agree that the resulting prior is far from perfect (see also Section 5, l326). Note that we settled for 3 curves, as using all 11 curves would have only partially fixed this issue, made this ‘peaking the prior’ exercise much more complicated, and in relative comparisons would likely negatively impact our MCMC baseline. In Section 4.2, we also compare to the original implementation using all 11 basis curves, to establish that LC-PFN is competitive on real learning curves, despite using only 3 basis curves.
>
> **The results could be more convincing if the authors can compare against more previous methods, instead of MCMC only. For instance, how about the one by Klein et al., 2017, which used Bayesian neural networks for learning curve prediction?**
>
> To the best of our knowledge, and based on a recent literature survey (Mohr and van Rijn 2022), Domhan et al. (2015) is the only work that previously considered Bayesian LCE requiring ONLY a single partial learning curve as input. LCNet (Klein et al., 2017), see also our discussion in the last paragraph of Section 2 (l94), in addition, requires the training hyperparameter settings as input. In fact, the Bayesian neural network used by Klein et al. (2017) only models the dependency of the prior on these training hyperparameters. This dependency must be meta-learned across multiple training runs. The only way we could compare to LCNet, in the current scope, would have been to remove this dependency (e.g., fix the input of the BNN to 0), but this would essentially reduce the BNN to a (gradient-based) MCMC method.
>
> If you are aware of any other prior art that should be included in our comparison, please let us know, and we would gladly add it.
>
>
> ### Regarding your questions / comments:
>
> **In Table 1, is the support for x integers, as it corresponds to $f_k(t | \theta_k)$  in the model?**
>
> Yes. To avoid confusion, we will update the table to use t instead of x here.
>
> **For the equation just below L132, what does < mean?**
>
> It simply means “less than”, comparing two real values (initial and final model performance), constraining the final performance to be better than the initial one (following Domhan et al., 2015).
>
> **Do you need to assume $t’>T$ in your LC-PFN model during training?**
>
> No, one could easily train an LC-PFN by censoring arbitrary points in the curves (vs right censoring only). The benefit is that the resulting model would also be able to do interpolation at the cost of requiring a slightly larger model/training time. Since our work is about extrapolation, we decided to train LC-PFN for this task specifically.

---

### Official Review · Reviewer_tPbz · 2023-07-07

**Soundness:** 3 good
**Presentation:** 3 good
**Contribution:** 3 good
**Rating:** 6
**Confidence:** 2

**Summary:**

The authors propose applying prior-data fitted NNs to learning curve extrapolation.

The authors demonstrate that this approach outperforms MCMC inference and is substantially faster.


**Strengths:**

The paper has a novel idea of applying approximate inference via meta-learning learning curves.

The method appears to work in experimental evaluation, but more experiments would make the paper stronger.

The manuscript (all but experimental section) is clear and relatively easy to follow.

The authors provide the code.



**Weaknesses:**

The paper contains only two experiments. One is based on synthetic data.

Table 1 has no error bars. Table 1 indicates that while the performance of MCMC is stable across the hyperparameters, PFN score is dependent on the network size indicating it might not generalize well to other problems.

While the difference in time is clearly significant, the differences in LL are rather small.

While the idea of applying the described method in the considered context is novel, much of this work is a combination of previous work by Domhan 2015 and Muller 2022.

The experimental section could have been written in a more clear way.

**Questions:**

Is poor performance of MCMC attributed to slow mixing or initializing at a poor point? What about MCMC inference initialized at MAP?

If MCMC has substantially more computational time will it outperform PFN? At the end of the day exact inference should outperform approximate inference.

---

> ### Author Rebuttal · Authors · 2023-08-10
>
> ### Regarding the weaknesses you raised
>
> **The paper contains only two experiments. One is based on synthetic data.**
>
> We evaluated LC-PFN in three different experimental setups (~ Section 4.1, 4.2, and 4.3). In Section 4.1, we evaluate LC-PFN (and MCMC) on samples of the prior. From a learning curve perspective, this is indeed ‘synthetic’ data. However, from a Bayesian perspective, this is arguably the best way of evaluating and comparing the ability of both methods to perform approximate inference without confounding factors. In Sections 4.2 and 4.3 we use real learning curve data from four different benchmark suites, training a wide variety of different architectures (e.g., MLP, CNN, RNN, Transformer) on a variety of different datasets (e.g., tabular, images, text, protein data) using a variety of different hyperparameter settings (e.g., optimizer, learning rate).
>
> **Table 1 has no error bars.**
>
> Table 1 in the paper does not present experimental results. We assume you mean Table 3? Our apologies, please find the updated table including the standard error in the PDF attached to the global rebuttal (as Table 1).
>
>
> **Table 1 indicates that while the performance of MCMC is stable across the hyperparameters, PFN score is dependent on the network size indicating it might not generalize well to other problems.**
>
> Table 3 only shows three configurations for each method. In total, we evaluated 27 PFN variants and 216 MCMC variants (see Table 2), and the results for all of these are shown in Figure 3 of the paper. Here, you can see that there is also variability amongst the MCMC configurations (red dots). For PFNs, variability is mostly determined by nb_data (how many curves the PFN was trained on) and to a lesser extent its size (``nlayers`` and ``emsize``). In fact, the figure shows that each of the 9 PFNs considered can be trained to outperform any of the MCMC configurations. While larger models do better (as also observed in other studies using transformers, e.g., Kaplan et al., 2020), this is generally viewed positively (e.g., the method can be scaled up with the availability of compute).
>
> **While the difference in time is clearly significant, the differences in LL are rather small.**
>
> Agreed, our main claim is that LC-PFN is faster without loss of quality.
>
> **While the idea of applying the described method in the considered context is novel, much of this work is a combination of previous work by Domhan 2015 and Muller 2022.**
>
> Agreed, our work essentially applies the method of Muller 2022 (PFNs) to the problem of Domhan 2015 (Bayesian LCE). However, given the novelty of the method, the relevance of the problem, and the magnitude of improvement (in terms of time complexity / practicality), we believe this to be a very significant contribution nonetheless; reviewer 3hos also seems to share this perspective. Also, we made several smaller, yet relevant contributions. In particular, we introduce a novel normalization procedure (see Appendix A), allowing both LC-PFN, and Domhan et al.’s (2015) MCMC to be applied to learning curves using a wide variety of different, possibly unbounded, performance metrics (not just maximizing a metric in [0,1], e.g., accuracy), including the popular “minimizing log loss”.
>
> **The experimental section could have been written in a more clear way.**
>
> Thank you for your feedback, we will improve the writing of this section for the final version but any concrete suggestions you have to further improve this section would be much appreciated.
>
> ### Regarding your questions (in reverse order):
>
> **If MCMC has substantially more computational time will it outperform PFN? At the end of the day exact inference should outperform approximate inference.**
>
> You are right, at least in theory and in the limit this should be the case (MCMC with finite chains is also approximate inference). However, the trends observed in Figure 3, in Section 4.1, seem to suggest only marginal improvements can be made by further increasing chain length. To further address your concern, we conducted extensive experiments the results of which can be found in the PDF attached to our global response. In summary, we discovered / fixed what we presume to be a bug in the original implementation (see global response for details), and ran experiments considering up to 50x longer chain lengths. The trends we observe (see Figure 1 in the rebuttal PDF), suggest that MCMC could eventually attain or even overtake the best LC-PFN. That being said, the best MCMC has approx. 25.000x longer runtimes than LC-PFN requires, MCMC yet not quite reach its performance, so outperforming it would require impractically long chains.
>
> **Is poor performance of MCMC attributed to slow mixing or initializing at a poor point? What about MCMC inference initialized at MAP?**
>
> This is a very good question. First, a few words on how we initialize the chain (also see Domhan et al., 2015, Section 3.1, page 4). Following Domhan et al. (2015), parameters for each of the basis curves are Least-Squared Estimates (LSE) obtained using the BFGS optimizer (Scipy implementation). Weights are initialized to be 1/K, where K(=3) is the number of basis curves. If this initial point violates the constraints imposed by the prior, a default starting point is used instead. Following your recommendation, we investigate the behavior of MCMC with different initializations including starting from the MAP estimate and the default value of the parameters. Figure 1 in the rebuttal PDF shows that the MAP does not improve over the original LSE implementation of Domhan et al. (2015). The fixed default initialization is worst for short chains, but best for very long chains (>20,000), suggesting that greedy initialization (LSE, MAP) may hurt performance on some curves.

---

> > ### Comment · Reviewer_tPbz · 2023-08-17
> >
> > Thank you for the detailed response. I maintain my score.

---

### Official Review · Reviewer_KctK · 2023-07-10

**Soundness:** 3 good
**Presentation:** 3 good
**Contribution:** 3 good
**Rating:** 7
**Confidence:** 4

**Summary:**

The authors used prior fitted networks to perform learning curve extrapolation. Crucially, they demonstrate that their method vastly outperforms approximate Bayesian inference via MCMC (both in terms of inference time and predictive log likelihood). Moreover, they demonstrate that the proposed approach outperforms heuristics for ending unpromising training runs commonly used in hyper-parameter optimization.

**Strengths:**

This paper prevents a super practical method that not only performs well *but* also runs fast, allowing it to be used in many real-time applications. The practicality of the method is extremely bolstered by the impressive suite of empirical experiments performed. Lastly, the paper was written well and was pretty easy to follow (though I have some comments).

**Weaknesses:**

The biggest weakness to me is that I don't think the authors spent time on the potential difficulty of using MCMC for this problem. Specifically, the prior, and the corresponding posterior, is constrained on some non-standard subset that prevents standard MCMC algorithms to be used. The geometry of this will heavily affect the performance of MCMC algorithms as samples that are proposed outside of the set will always be rejected requiring substantial tuning. That being said, I am a little concerned with the number of samples the chain was run: 4,000 samples seems too low for this problem.

**Questions:**

Below I will list some comments
1. In figure 1 there is a black dashed line but I can't find in the text what it represents. Without prior knowledge, I thought that maybe it was the true log posterior predictive distribution, which would make it seem like MCMC is outperforming the proposed method. Also, error bars here would be great.
2. For all results, it would help if it was made clear if higher values are better or the converse.
3. For Fig. 4, errors bar would be very informative, especially given the range of values. I suggest plotting 20% and 80% quantiles.
4. In Fig 5., it isn't clear what the error bars are, i.e. standard error, quantiles, etc.

**Limitations:**

The authors did a good job explaining the limitation of the methods.

---

> ### Author Rebuttal · Authors · 2023-08-10
>
> ### Regarding the weaknesses you raised:
>
> **The biggest weakness to me is that I don't think the authors spent time on the potential difficulty of using MCMC for this problem. Specifically, the prior, and the corresponding posterior, is constrained on some non-standard subset that prevents standard MCMC algorithms to be used ….**
>
> We agree that Bayesian LCE is a non-trivial problem for MCMC. In fact, this was an important motivation of LC-PFN. For our MCMC baseline, we relied on previous work (Domhan et al., 2015) to make MCMC work reasonably in this challenging setting. While we were forced to make minor changes to the setup, in particular to the prior, we made considerable efforts to retain the competitiveness of the original implementation. Furthermore, we also included the original MCMC implementation as a baseline in our experiments in Section 4.2. To further address your concern, we conducted extensive experiments the results of which can be found in the PDF attached to our global response. First, it is important to note that our original results in Figure 3 in the paper are averages across 10,000 curves. When comparing performances on individual curves, shown in Figure 2 (left, red dots) in the attached PDF, we observe that for the majority of the curves, the best LC-PFN and MCMC perform similarly (i.e., are near the diagonal), and inferior average performance can be largely attributed to ‘failures’ of MCMC on a small fraction of the curves, the ones that are in the lower-right region of Figure 2 (from the rebuttal PDF). After analyzing these failure cases of MCMC, we noticed behavior inconsistent with what was described by Domhan et al. (2015) and fixed the bug we thereby identified as discussed in the global response. Doing this and running MCMC much longer, as you suggested (more below), reduces these failure cases considerably (see Figure 2 (left to right, red vs green dots)).
>
> **I am a little concerned with the number of samples the chain was run: 4,000 samples seems too low for this problem.**
>
> We understand your concern. It is worth noting that Domhan et al. (2015) only used 2,000 samples, which motivated our choice of 4,000 samples as an upper limit. Also, please note that Domhan’s implementation (which we use) uses an ensemble sampler (EMCEE) with 100 workers, and the number of 4,000 samples is actually for each worker, which results in 400,000 samples overall. The trend we observed in Figure 3 of our paper is that while increasing chain length indeed improves performance, the slow rate of improvement suggests that impractically long chains would be required to attain the performance of even the smallest PFN. However, after fixing the bug mentioned above in Domhan’s original implementation, improvement no longer stagnates, and results running MCMC for up to 100,000 samples per worker (i.e., a total of 10M samples) suggest that the gap in performance will close with even more samples (see Figure 1 in the rebuttal PDF, red line). That being said, MCMC still does not reach the average performance of the best LC-PFN, while taking approximately 25,000 times as long.
>
> ### Regarding your questions / comments:
>
> **In figure 1 there is a black dashed line but I can't find in the text what it represents. Without prior knowledge, I thought that maybe it was the true log posterior predictive distribution, which would make it seem like MCMC is outperforming the proposed method. Also, error bars here would be great.**
>
> Unless we are mistaken, there is no black dashed line in Figure 1 of the paper. The red/blue dashed lines in Figure 1 represent the 5 and 95 percentiles of the PPD (so the shaded area ~ 90% CI) as inferred using MCMC/LC-PFN, respectively. We will update the legend of Figure 1 to clarify this. We assume you meant Figure 3 instead. The black dashed line in Figure 3 represents the highest log score attained by any MCMC variant (higher is better). Thus, this does not imply that MCMC is outperforming the proposed method but the opposite. We will update the legend / caption of Figure 3 to clarify this.
>
> **For all results, it would help if it was made clear if higher values are better or the converse.**
>
> Definitely, thank you for the suggestion, FYI:
> - Figure 3 / Table 3:
>     - Higher is better for log score (average LL)
>     - Lower is better for average runtime
> - Figure 4: Lower is better (average rank)
> - Figure 5: Lower is better (average regret)
> We will clarify this in the captions.
>
> **For Fig. 4, errors bar would be very informative, especially given the range of values. I suggest plotting 20% and 80% quantiles.**
>
> Agreed. However, as Figure 4 shows averages of ranks amongst 3 methods, we feel quantiles (which will be one of the integers 1, 2, or 3) would not be very informative. We propose to use +-1 standard error (SE) instead.
>
> **In Fig 5., it isn't clear what the error bars are, i.e. standard error, quantiles, etc.**
>
> The shaded area corresponds to +- 1 standard error. We will update the caption to clarify this, thanks!

---

### Author Rebuttal · Authors · 2023-08-10

In this global response, we would like to thank all reviewers for their constructive feedback. We are glad that our work was generally well-received and we address specific concerns / questions raised by the reviewers in our individual responses. Multiple reviewers expressed some concerns about our MCMC baseline. To support our response, we conducted extensive experiments that we discuss in the following; accompanying figures are found in the attached PDF.

**Improving baseline implementation:** We analyzed cases where MCMC performed poorly in order to determine any potential causes of its non-convergence to the optimum. In these analyses, we noticed behavior inconsistent with what was described by Domhan et al. (2015). Specifically, the hard constraints (monotonicity and range) were checked for every basis curve ($f_k$), rather than on the combination ($f_{comb}$), as described in their paper (see Domhan et al., 2015, Equation 5). While we assume this was intended as an optimization in their code, it is not equivalent (i.e., a combination can satisfy these constraints, while individual curves do not) and this is exactly what was happening for the curves MCMC performed worst on. After fixing this (which we would call a bug) in their original implementation, we re-ran the fixed version of the controlled experiment (Section 3). This yielded results much more in line with what one would expect from MCMC for higher sample sizes: while it takes a long time, it finds solutions as good as those by LC-PFN (see Figure 1 in the rebuttal PDF). It is crucial to emphasize that despite these improved results, our main claim about LC-PFN obtaining many orders of speedup without loss of quality w.r.t. MCMC remains valid.


**Ablating MCMC w.r.t the initialization and the number of samples:** We ran experiments to assess the effect of the initialization strategies, including the original least-square estimate (LSE), the Maximum a posteriori (MAP) estimate, and a fixed default value of the parameters. Furthermore, we ran experiments on considerably larger sample sizes (up to 100,000 samples). As shown in Figure 1 in the attached PDF, LC-PFN still consistently outperforms MCMC in this ablation.

---

### Decision · Program_Chairs · 2023-09-21

**Decision:**

Accept (poster)

**Comment:**

This paper applies prior fitted networks (PFNs) to learning curve extrapolation. Predicting learning curve for early stopping is an important research topic in AutoML. This work applies PFNs to improve the inference of the Bayesian model proposed in Domhan et al. (2015). It results in better prediction and significantly higher computation efficiency.

The main idea is an application of PFNs to the model of Domhan et al. (2015) with some modification on the prior although some reviewers still consider this novel application in learning curve extrapolation as a good contribution.

Reviewers consider it as a highly practical method with good performance. It is also evaluated on a variety of real-data experiments.

Some reviewers have concerns in the experiments, including (1) the Bayesian model with only 3 basis curves maybe too simple to capture all learning curves (2) comparison with additional existing works. The authors' rebuttal provided explanation on the model choice and why the additional work is not applicable in the scope of the paper.